# IMPROVING THE SENSITIVITY OF BACKDOOR DETECTORS VIA CLASS SUBSPACE ORTHOGONALIZATION

## ABSTRACT

Most post-training backdoor detection methods rely on attacked models exhibiting extreme outlier detection statistics for the target class of an attack, compared to non-target classes. However, these approaches may fail: (1) when some (non-target) classes are easily discriminable from all others, in which case they may *naturally* achieve extreme detection statistics (e.g., decision confidence); and (2) when the backdoor is subtle, i.e., with its features weak relative to intrinsic class-discriminative features. A key observation is that the backdoor target class has contributions to its detection statistic from both the backdoor trigger *and* from its intrinsic features, whereas non-target classes *only* have contributions from their intrinsic features. To achieve more sensitive detectors, we thus propose to *suppress* intrinsic features while optimizing the detection statistic for a given class. For non-target classes, such suppression will drastically reduce the achievable statistic, whereas for the target class the (significant) contribution from the backdoor trigger remains. In practice, we formulate a constrained optimization problem, leveraging a small set of clean examples from a given class, and optimizing the detection statistic while orthogonalizing with respect to the class's intrinsic features. We dub this approach "class subspace orthogonalization" (CSO). CSO can be "plug-and-play" applied to a wide variety of existing detectors. We demonstrate its effectiveness in improving several well-known detectors, comparing with a variety of baseline detectors, against a variety of attacks, on the CIFAR-10, GTSRB, and TinyImageNet domains. Moreover, to make the detection problem even more challenging, we also evaluate against a novel mixed clean/dirty-label poisoning attack that is more surgical and harder to detect than traditional dirty-label attacks. Finally, we evaluate CSO against an adaptive attack designed to defeat it, with promising detection results.

## 1 INTRODUCTION

Deep learning relies on large labeled training sets, making it vulnerable to backdoor attacks. Here, the adversary manipulates the training process of a deep neural network (DNN) classifier, typically via training set poisoning, so that the model learns to associate the attacker's chosen trigger pattern with its designated target class. A backdoored model will perform normally on clean inputs, but will **misclassify** inputs containing the trigger to the target class. Because the poisoned fraction of the training set is small and because the trigger can be **innocuous** (visually subtle), such attacks can evade conventional validation, posing security risks for real-world deployment.

To address this threat, a variety of backdoor defenses have been proposed. Mitigation methods can be applied pre-training, to sanitize the training set of poisoned samples, or post-training, with model fine-tuning on clean samples used to purge a backdoor mapping from the model's weights. The focus of this work, on the other hand, is *post-training backdoor detection*. Here, the defender *only* has access to the trained model and would like to infer whether it has been backdoored and, if so, the target class of the attack. The post-training scenario is of great significance, considering an entity (e.g., a government agency) may purchase a pre-trained AI, with no rights to access the training set. Consider also a viable small business model wherein a company fine-tunes a (possibly backdoored) pre-trained foundation model for an end-user application.

Post-training detection – without any access to the possibly poisoned training set – is quite challenging since in this case the only "evidence" for a backdoor is its imprinting on the trained weights of the network. Some of the earliest post-training methods (e.g., (Wang et al. (2019); Xiang et al. (2022))) are *reverse-engineering* detectors that assume knowledge of the mechanism (e.g., patch (Gu et al. (2019)), blend, additive) by which the backdoor trigger pattern is encoded into a sample. These methods are effective when the mechanism assumed by the defense is the one actually used by the attacker, but they may fail under mechanism mismatch. More recent post-training detectors do not assume knowledge of the attacker's mechanism. (Wang et al. (2022a; 2023); Xu et al. (2024)) aim to *learn* a (UNet) model capturing the attacker's mechanism, while MMBD (Wang et al. (2024)) hypothesizes that, *irrespective of* the mechanism used, the model tends to overfit to the backdoor trigger. While these previous works, e.g., (Wang et al. (2019); Xiang et al. (2022); Wang et al. (2022a; 2023); Xu et al. (2024)), differ in their assumptions, the *commonality* of these methods is that they are based on solving an optimization problem that yields a detection statistic for each class, with detections made when one class's statistic is a significant outlier. In fact, these methods all try to maximize the average class posterior for a putative target class, measured over a clean set of samples originating from other classes. MMBD likewise maximizes class margin – the difference between the logit for a putative target class and the largest logit among the remaining classes – and does not exploit a clean set of samples.

While these approaches have demonstrated strong results, they can fail in two important settings. First, when the poisoning rate is quite low, but still sufficient to implant an effective backdoor, the trigger's "signal" may be too weak to yield a significant outlier detection statistic, resulting in a missed detection. Second, when certain classes have particularly strong or unique intrinsic features, these features may produce unusually large detection statistics even in backdoor-free models, resulting in false positives. Moreover, in backdoored models, strong intrinsic features for non-target classes may yield detection statistics that rival that for the target class. In such case, the target class's statistic will not be a significant outlier, again resulting in a missed detection.

A key observation is that the backdoor target class has contributions to its detection statistic from both the backdoor trigger *and* from its intrinsic features, whereas non-target classes *only* have contributions from their intrinsic features. Thus, to achieve more sensitive detectors, we propose to *suppress* intrinsic features while optimizing the detection statistic for a given class. For non-target classes, such suppression will drastically reduce the achievable statistic (e.g., class margin), whereas for the backdoor target class the (significant) contribution from the backdoor trigger will remain. To achieve such suppression, we propose Class Subspace Orthogonalization (CSO), a general optimization approach wherein the detector's loss objective for a given class is altered to include a term penalizing feature directions that are not *orthogonal to* the class's intrinsic features. CSO thus guides the detector to search in feature directions orthogonal to intrinsic features of the putative target class, i.e., those where the backdoor trigger's features are more likely to "reside". Integrating CSO into the optimization process of existing detectors helps the detector to be both more sensitive to backdoor patterns and more robust against intrinsic feature interference. Importantly, CSO is *detector-agnostic*, i.e., it can be "plug and play" applied (as seen in the sequel) to a wide range of post-training detection methods, without requiring changes to their core algorithms (excepting the modification of the detector's loss objective). Moreover, CSO only requires a small set of clean samples for each class, which aligns well with many existing detectors that already make use of clean data.

Also, to more stringently test the effectiveness of CSO and baseline methods for backdoor detection, we also introduce a novel, mixed dirty/clean label X-to-X poisoning attack that minimizes *collateral damage* (Xiang et al. (2022)), i.e., the misclassification of *non-source class*, trigger-laden samples to the target class of the attack. We note that such misclassifications are not *intended* by the attacker. In (Xiang et al. (2022)), it was shown that collateral damage rates are quite high for traditional dirty label 1-to-1 (a single (source, target) class pair) attacks. To reduce these unintended misclassifications, we propose a *mixed* dirty/clean label attack, with trigger-laden samples from designated source classes (dirty-) labeled to the target class of the attack, and with trigger-laden samples from *non*-source classes (clean-) labeled to their class of origin. In addition to yielding a more "surgical" backdoor mapping, this novel attack is also substantially harder to detect than traditional dirty label attacks, as will be seen, which makes it a good, subtle attack against which to assess both CSO and existing baseline detectors.

The contributions of this paper are four-fold: (1) Our primary contribution is CSO, a novel framework that guides the detector's optimization process toward backdoor signal directions while suppressing intrinsic class-specific signals that could otherwise dominate detection statistics. CSO is flexible and can be seamlessly integrated with existing backdoor detectors. (2) As a secondary contribution, we propose a mixed clean/dirty-label attack that applies clean-label poisoning to non-source classes. This results in an attack that is harder to detect than traditional dirty label attacks, which makes it a good challenge for both baseline and new (CSO) detectors. (3) We conduct extensive experiments on multiple benchmark datasets, demonstrating CSO's effectiveness when combined with SOTA detectors and its robustness against adaptive attacks. (4) We comprehensively evaluate the mixed-label attack across multiple backdoor types, datasets, and network architectures, showing its ability to significantly degrade a wide range of defenses.

The rest of the paper is as follows. Section 2 develops our CSO detector. Section 3 gives experimental results. Section 4 discusses related work. Finally, Section 5 gives conclusions and directions for future work. Additional details and ablation studies are given in the appendix.

## 2 METHODOLOGY

### 2.1 THREAT MODEL

**Attacker's Goal and Capabilities.** In our mixed-label attack, the adversary has the same capabilities as in prior work (Gu et al. (2019); Wang et al. (2019); Xiang et al. (2022)): the ability to (passively) poison the training set, but no access to the training process or model implementation.

**Defender's Goal and Capabilities.** We consider a post-training defense setting, where the classifier has already been trained and the original training set is unavailable. The defender's goal is to determine whether the classifier has been backdoor-attacked and to infer the attack's target class. Following prior defenses (Wang et al. (2019); Xiang et al. (2022); Wang et al. (2023); Yang et al. (2024); Xu et al. (2024)), we assume that the defender can independently collect a small, clean dataset containing samples from all classes in the domain.

### 2.2 CLASS SUBSPACE ORTHOGONALIZATION

Generally, CSO contains *two* steps: (1) identifying the intrinsic features for each putative target class and (2) suppressing these intrinsic features during detector optimization by enforcing orthogonality to the class's intrinsic subspace.

#### 2.2.1 MOTIVATION FOR THE CASE OF A MAXIMUM LOGIT DETECTOR

Consider a $K$-class problem, working in the space of augmented feature vectors $\boldsymbol{y}^T = (\boldsymbol{x}^T, 1)$. Suppose the class set is $\mathcal{K}$, and for each class, $k \in \mathcal{K}$, there is a linear discriminant function $g_k(\boldsymbol{y}) = \boldsymbol{w}_k^T \boldsymbol{y}$. The classifier employs a winner-take-all decision rule: $\hat{c}(\boldsymbol{y}) = \arg\max_j g_j(\boldsymbol{y})$. For class $k$, the clean training set is $\mathcal{Y}_k = \{\boldsymbol{y}_1^{(k)}, \ldots, \boldsymbol{y}_{N_k}^{(k)}\}$. Suppose the training set for source class $s$ is dirty-label backdoored with target class $t$. For non-target classes $k \neq t$, we assume the weight vector can be expressed in the form $\boldsymbol{w}_k = \sum_{i=1}^{N_k} \gamma_i^{(k)} \boldsymbol{y}_i^{(k)}$, where $\gamma_i^{(k)} \geq 0, \forall i$, i.e., it is a non-negative linear combination of the training vectors. This holds, for example, if the linear classifier is a multi-class linear Support Vector Machine or if it is learned via a multi-class extension of the Perceptron algorithm. For the target class $t$, we write $\boldsymbol{w}_t = \sum_{i=1}^{N_t} \gamma_i^{(t)} \boldsymbol{y}_i^{(t)} + \alpha \boldsymbol{b}^{(t)}$, where $\gamma_i^{(t)} \geq 0, \forall i, \alpha > 0$, and $\boldsymbol{b}^{(t)}$ is a sample containing the backdoor pattern. We will suppose that $\boldsymbol{b}^{(t)} \notin \text{span}(\mathcal{Y}_t)$. This is reasonable, given that the backdoor trigger *modifies* samples from class $s \neq t$. For example, in an *additive* attack mechanism, $\boldsymbol{b}^{(t)} = \boldsymbol{y}^{(s)} + \boldsymbol{b}$, $\boldsymbol{y}^{(s)}$ a source-class sample and $\boldsymbol{b}$ the backdoor pattern. Clearly, in general, $\boldsymbol{y}^{(s)} \notin \text{span}(\mathcal{Y}_t)$ and, thus, $\boldsymbol{b}^{(t)} \notin \text{span}(\mathcal{Y}_t)$.

**Baseline Maximum-Logit Backdoor Detection (MLBD).** A maximum-logit detector determines, for each class $k$:

$$l_k = \max_{\boldsymbol{y}} \boldsymbol{w}_k^\top \boldsymbol{y} \quad \text{s.t.} \quad \|\boldsymbol{y}\|_2 = 1, \tag{1}$$

with the constraint needed to make the problem well-posed. The detector then flags a backdoor if $l_{k^*} = \max_k l_k$ is an outlier. However, if a non-target class $\tilde{t} \neq t$ has strong intrinsic features such that $l_{\tilde{t}} \approx l_t$ or even $l_{\tilde{t}} > l_t$, $l_t$ may not be an outlier, leading to a missed detection.

**Orthogonalized MLBD.** To resolve this issue, we modify the optimization by adding class-subspace orthogonality constraints:

$$l_k = \max_{\boldsymbol{y}} \boldsymbol{w}_k^\top \boldsymbol{y} \quad \text{s.t.} \quad \|\boldsymbol{y}\|_2 = 1, \quad \boldsymbol{y}^\top \boldsymbol{y}_i^{(k)} = 0, \ i = 1, \ldots, N_k. \tag{2}$$

That is, we must have: $\boldsymbol{y} \perp \text{span}(\mathcal{Y}_k)$. For $k \neq t$, since $\boldsymbol{w}_k \in \text{span}(\mathcal{Y}_k)$, we have $\boldsymbol{w}_k^\top \boldsymbol{y} = 0$ for all feasible $\boldsymbol{y}$, yielding $l_k = 0$. For $k = t$, we can write $\boldsymbol{w}_t = \boldsymbol{w}_t^{\text{benign}} + \boldsymbol{b}_\perp^{(t)}$, where $\boldsymbol{w}_t^{\text{benign}} \in \text{span}(\mathcal{Y}_t)$, and $\boldsymbol{b}_\perp^{(t)}$ is the component of $\alpha\boldsymbol{b}^{(t)}$ orthogonal to $\text{span}(\mathcal{Y}_t)$. The orthogonality constraint removes $\boldsymbol{w}_t^{\text{benign}}$, leaving:

$$l_t = \max_{\boldsymbol{y}} \boldsymbol{b}_\perp^{(t)\top} \boldsymbol{y} \quad \text{s.t.} \quad \|\boldsymbol{y}\|_2 = 1, \tag{3}$$

whose solution is $\boldsymbol{y}^* = \boldsymbol{b}_\perp^{(t)}/\|\boldsymbol{b}_\perp^{(t)}\|$ with $l_t = \|\boldsymbol{b}_\perp^{(t)}\| > 0$. Thus, via orthogonalization, all $l_k$ vanish for $k \neq t$ while $l_t$ remains positive, making $l_t$ a clear outlier. This suggests orthogonalization can be the basis for improved detection sensitivity.

### 2.2.2 IDENTIFYING INTRINSIC, CLASS-SPECIFIC FEATURES

For purposes of backdoor detection, let the DNN classifier be $f(\cdot) = S_b \circ S_a(\cdot)$, where $S_a$ denotes the feature extractor (e.g., convolutional layers) and $S_b$ performs back-end classification. For an input $\boldsymbol{x}$ in the input space $\mathcal{X}$, the internal feature vector is $S_a(\boldsymbol{x}) \in \mathbb{R}^d$. Inspired by feature decoupling methods (Wang et al. (2022a; 2023); Xu et al. (2024), Wang et al. (2019)), for each class $k$, we aim to learn a soft mask $\boldsymbol{v}_k \in [0,1]^d$ that identifies its intrinsic feature subspace. Given a small clean set for class $k$, $\mathcal{D}_k = \{\boldsymbol{x}_i^{(k)}, i = 1, \ldots, M_k\}$, we choose $\boldsymbol{v}_k$ to (simultaneously) minimize classifier loss on the intrinsic features and maximize classifier loss on the complement set:

$$\min_{\boldsymbol{v}_k} \sum_{\boldsymbol{x}^{(k)} \in D_k} \left[ \mathcal{L}(S_b(S_a(\boldsymbol{x}^{(k)}) \odot \boldsymbol{v}_k), k) - \mathcal{L}(S_b(S_a(\boldsymbol{x}^{(k)}) \odot (1 - \boldsymbol{v}_k)), k) \right], \tag{4}$$

where $\odot$ is the element-wise (Hadamard) product, and $\mathcal{L}$ is, e.g., the cross-entropy loss. This produces an intrinsic subspace for each class $k$: $\text{span}(\boldsymbol{v}_k \odot S_a(\boldsymbol{x}^{(k)}) : \boldsymbol{x}^{(k)} \in \mathcal{D}_k)$. Note that, unlike existing feature decoupling methods that use soft-masking in activation space (Wang et al. (2022a; 2023); Xu et al. (2024)), we are estimating a *class-specific* mask. This is both reasonable and necessary since the features that are class-discriminating will in general be class-specific.

### 2.2.3 ORTHOGONALIZATION IN DETECTOR OPTIMIZATION

Let $\boldsymbol{\Theta}$ be the variables over which the detector optimizes (e.g., the DNN's input, or parameters specifying how a putative backdoor trigger is incorporated into a sample, in the case of a reverse engineering based detector). We write $\boldsymbol{z}(\boldsymbol{\Theta}; \boldsymbol{x})$ to represent a putative backdoor-triggered sample induced by $\boldsymbol{\Theta}$ (potentially acting on an input sample, $\boldsymbol{x}$), with $S_a(\boldsymbol{z}(\boldsymbol{\Theta}; \boldsymbol{x}))$ its internal features.

For each putative target class $t$, exploiting the learned class-specific mask $\boldsymbol{v}_t$, to effectively orthogonalize, we penalize positive correlations between $S_a(\boldsymbol{z}(\boldsymbol{\Theta}))$ and the intrinsic subspace for class $t$. That is, we define the CSO penalty for $\boldsymbol{z}(\boldsymbol{\Theta}; \boldsymbol{x})$ as the average rectified cosine similarity:

$$C_t(\boldsymbol{z}(\boldsymbol{\Theta}; \boldsymbol{x})) := \frac{1}{|D_t|} \sum_{\boldsymbol{x}^{(t)} \in D_t} \text{ReLU}\left( \frac{\langle S_a(\boldsymbol{z}(\boldsymbol{\Theta}; \boldsymbol{x})), \boldsymbol{v}_t \odot S_a(\boldsymbol{x}^{(t)}) \rangle}{\|S_a(\boldsymbol{z}(\boldsymbol{\Theta}; \boldsymbol{x}))\| \|\boldsymbol{v}_t \odot S_a(\boldsymbol{x}^{(t)})\|} \right), \tag{5}$$

with inner (dot) product $\langle \cdot, \cdot \rangle$. This novel penalty will be used to discourage alignment with intrinsic features of a putative target class, pushing the detector's search into subspaces more likely to contain backdoor-related components, and with the ReLU operation leaving *anti-correlated* directions unaffected.

### 2.3 DETECTORS + CSO

Let $\mathcal{J}_t(\cdot)$ denote the objective function of a given detector for putative target class $t$. We next develop CSO variants of several well-known detectors.

### 2.3.1 MMBD-CSO

MMBD (Wang et al. (2024)) exploits an empirically observed backdoor overfitting phenomenon, searching for an input that maximizes the *margin*, i.e., the (non-negative) difference between the logit for putative target class $t$ and the second-largest logit. In this case, since the detector directly optimizes over the input space, we have $\boldsymbol{z}(\boldsymbol{\Theta}; \boldsymbol{x}) = \boldsymbol{z} \in \mathcal{X}$. Incorporating the CSO penalty term, we have the following MMBD-CSO loss objective:

$$\mathcal{J}_t(\boldsymbol{z}) = -\big[g_t(\boldsymbol{z}) - \max_{k \in \mathcal{K} \backslash t} g_k(\boldsymbol{z})\big] + \lambda C_t(\boldsymbol{z}), \tag{6}$$

with $\lambda$ giving the relative weight to the CSO penalty. The detection statistic in this case is the achieved maximum margin. Note that the original MMBD approach had no way to benefit from any available clean samples. MMBD-CSO exploits clean samples, and as will be seen, significantly improves on the detection performance of MMBD.

We also consider Maximum Logit Backdoor Detection (MLBD)-CSO, with the margin term replaced just by the logit:

$$\mathcal{J}_t(\boldsymbol{z}) = -g_t(\boldsymbol{z}) + \lambda C_t(\boldsymbol{z}). \tag{7}$$

In this case the detection statistic is the achieved maximum logit.

### 2.3.2 NC-CSO

Neural Cleanse (Wang et al. (2019)) assumes a *blended* attack mechanism, defined by a *spatial* soft mask $\boldsymbol{m}_{\mathrm{NC}} \in [0, 1]^{\dim(\mathcal{X})}$ and trigger pattern $\boldsymbol{p} \in \mathcal{X}$, i.e., $\Theta = \{\boldsymbol{m}_{\mathrm{NC}}, \boldsymbol{p}\}$ and $\boldsymbol{z}(\boldsymbol{\Theta}; \boldsymbol{x}) = (1 - \boldsymbol{m}_{\mathrm{NC}}) \odot \boldsymbol{x} + \boldsymbol{m}_{\mathrm{NC}} \odot \boldsymbol{p}$. The Neural Cleanse detector jointly chooses the spatial mask and pattern to effectively maximize the fraction of non-target class clean samples induced, by the blending operation, to be classified to the putative target class, $t$. With the incorporation of a CSO penalty, the NC-CSO loss objective becomes:

$$\mathcal{J}_t(\{\boldsymbol{m}_{\mathrm{NC}}, \boldsymbol{p}\}) = \sum_{\boldsymbol{x}^{(k)} \in \mathcal{D}_k, \, k \neq t} \mathcal{L}\Big(f\big((1 - \boldsymbol{m}_{\mathrm{NC}}) \odot \boldsymbol{x}^{(k)} + \boldsymbol{m}_{\mathrm{NC}} \odot \boldsymbol{p}\big), t\Big) + r^{\star}$$
$$+ \lambda \sum_{\boldsymbol{x}^{(k)} \in \mathcal{D}_k, \, k \neq t} C_t\Big((1 - \boldsymbol{m}_{\mathrm{NC}}) \odot \boldsymbol{x}^{(k)} + \boldsymbol{m}_{\mathrm{NC}} \odot \boldsymbol{p}\Big). \tag{8}$$

where $\mathcal{L}$ is the cross entropy loss and $r^{\star}$ is a regularization term (e.g., to encourage small trigger size and small mask size). So, NC-CSO soft-masks *both* the input and the embedded ($S_a$) feature space. The soft-masking of the input (image) estimates the *spatial support* of the backdoor trigger, while soft-masking in activation space estimates the class's intrinsic features, defined on this spatially localized support. These two feature maskings are, thus, complementary, and both are necessary within NC-CSO. Here, the detection statistic is the estimated mask norm $\|\boldsymbol{m}_{\mathrm{NC}}\|$, where a smaller mask norm serves as evidence of a potential backdoor target class.

### 2.3.3 PT-RED-CSO

PT-RED (Xiang et al. (2022)) seeks the smallest additive perturbation $\boldsymbol{p} \in \mathcal{X}$ that causes most inputs from a putative source class $s$ to be misclassified to a putative target class $t$, i.e., in this case $\Theta = \{\boldsymbol{p}\}$ and $\boldsymbol{z}(\boldsymbol{\Theta}; \boldsymbol{x}) = \boldsymbol{x} + \boldsymbol{p}$. With a CSO penalty incorporated, the PT-RED-CSO loss objective becomes:

$$\mathcal{J}_t(\boldsymbol{p}) = \sum_{\boldsymbol{x}^{(s)} \in \mathcal{D}_s} \mathcal{L}\Big(f\big(\boldsymbol{x}^{(s)} + \boldsymbol{p}\big), t\Big) + \lambda \sum_{\boldsymbol{x}^{(s)} \in \mathcal{D}_s} C_t(\boldsymbol{x}^{(s)} + \boldsymbol{p}), \tag{9}$$

with $\mathcal{L}$ the cross entropy loss[1]. The detection statistic in this case is the perturbation norm $\|\boldsymbol{p}\|$, with smaller norms indicating higher likelihood of a backdoor mapping from $s \rightarrow t$.

Note that these 4 CSO variants differ both in the detector objective function and in the variables being optimized over. Thus, these 4 variants give a good idea about the "plug-and-play" generality of the CSO framework. More generally, so long as an existing detector is optimizing a detection statistic that relies on intrinsic feature contributions, CSO should in principle be applicable to this existing detector. The detection benefits of such inclusion will be seen shortly.

---

[1] Alternatively, PT-RED can apply an additive perturbation in activation space, see (Xiang et al. (2022)).

## 2.4 MIXED LABEL ATTACK

Next, we introduce a novel, stealthy attack that will be used in our experiments as a significant detection challenge, both for CSO detection methods and for the baseline methods with which we will compare. In an X-to-X attack, let the attacker's poisoned source class set be $\mathcal{S} \subseteq \mathcal{K}$ and the target class set be $\mathcal{T} \subseteq \mathcal{K}$. In a traditional dirty-label attack, the attacker mislabels a subset of samples from $\mathcal{S}$ to $\mathcal{T}$ and allows the model to be trained as usual. However, in our approach, the attacker performs an additional step by *also* embedding the backdoor trigger into samples from $\mathcal{K} \setminus (\mathcal{S} \cup \mathcal{T})$ *while keeping their original labels intact.* This teaches the model to learn to *only* misclassify to $\mathcal{T}$ when the backdoor trigger is applied to samples from $\mathcal{S}$, i.e., so that there is little to no collateral damage.

To characterize this attack, we introduce the following metrics: 1) Dirty Poisoning Rate (DPR): the number of poisoned samples from $\mathcal{S}$ mislabeled to $\mathcal{T}$, divided by the total number of training samples; 2) Clean Poisoning Rate (CPR): the number of clean-label poisoned samples divided by the total number of training samples; 3) Overall Poisoning Rate (OPR): the sum of DPR and CPR. In traditional dirty-label attacks, OPR = DPR. In contrast, we introduce additional poisoned but correctly labeled samples, making DPR < OPR. Similar to other attacks, at modest poisoning rates, our attack achieves i) a high attack success rate on samples from the intended source classes when the backdoor trigger is applied and ii) minimal degradation in clean test accuracy. However, unlike traditional attacks, our attack also greatly reduces collateral damage. Moreover, as will be seen, this attack is much harder to detect than conventional dirty-label poisoning and, thus, represents a good challenge, both for CSO and baseline detection methods.

## 3 EXPERIMENTAL RESULTS

### 3.1 EXPERIMENT SETUP

**Dataset and models.** We experiment on three benchmark datasets: CIFAR-10 (Krizhevsky & Hinton (2009)), GTSRB (Houben et al. (2013)), and (a subset of) TinyImageNet (Le & Yang (2015)), containing 40 classes. We evaluate our methods for ResNet-18 (He et al. (2016)) , PreActResNet-18 (Yu et al. (2018)), VGG-16 (Simonyan & Zisserman (2015)), and ViT (Dosovitskiy et al. (2021)) on CIFAR-10, MobileNet (Howard et al. (2017)) on GTSRB, and ResNet-34 on TinyImageNet. Results on VGG-16 and ViT are reported in Appendix A.6. We randomly select 10 clean training samples per class for use by our CSO variants. Please see Appendix A.1.1 for more dataset details.

**Backdoor Attacks.** We assess against thirteen representative and advanced backdoor attacks: (1) BadNets (Gu et al. (2019)), (2) chessboard (Xiang et al. (2022)), (3) 1-pixel (Tran et al. (2018)), (4) blend (Wang et al. (2019)), (5) WaNet (Nguyen & Tran (2021)), (6) input-aware (dubbed 'IAD') (Nguyen & Tran (2020)), (7) label-consistent (dubbed 'LC') (Turner et al. (2019)), (8) Bpp (Wang et al. (2022b)), (9) Refool (Liu et al. (2020)), (10) Narcissus (Zeng et al. (2023)), (11) SIG (Barni et al. (2019)), (12) Bypass (Shokri et al. (2020)) and (13) DRUPE (Tao et al. (2024)). Attacks (9)-(13) are evaluated in Appendix A.5. LC is not reported on TinyImageNet as the attack fails despite high poisoning rates. For the mixed dirty/clean-label attack (which we denote by 'ML' (mixed labeling) in the sequel), we consider both one-to-one and multi-to-one cases on BadNets, chessboard, 1-pixel and blend. CPR was set equal to DPR, *i.e.* a very modest amount of clean label poisoning – e.g., for BadNet both DPR and CPR were 0.1%. We experimented under a *lowest* feasible poisoning rate scenario. That is, the adversary chooses the *minimum* poisoning rate needed to achieve a high attack success rate. *This makes the detection problem most challenging.* Detailed attack settings—including the poisoning rate (PR), attack success rate (ASR), clean test accuracy (ACC), and, for one-to-one and multi-to-one attacks, collateral damage (CD)—are reported in Appendix A.1.3.

**Backdoor Defenses.** We tested five well-known backdoor defense methods as baselines, i.e., Neural Cleanse (NC) (Wang et al. (2019)), PT-RED (Xiang et al. (2022)), MMBD (Wang et al. (2024)), UNICORN (Wang et al. (2023)), and BTI-DBF (Xu et al. (2024)), compared against CSO variants of NC, PT-RED, MMBD, and MLBD. We closely adhered to the original implementations of all baseline methods. However, BTI-DBF always detects models as poisoned, that is, it gives 100% false positives on clean models. In order to reduce false positives, we endow BTI-DBF with a detection rule, applying a threshold to their maximum class proportion measure. We report under two thresholds: $1/|\mathcal{K}|$ (favoring backdoor detection) and 1.0 (minimizing false positives), respectively denoted BTI-DBF and BTI-DBF-2. UNICORN was excluded from the TinyImageNet evaluation due

to prohibitive runtime. Additional implementation details for all baselines are provided in Appendix A.1.2. For the CSO variants, we chose $S_a()$ as the last convolution layer. Justification for this choice is given in Appendix A.2.1. We set $\lambda$ to achieve approximate balance between the detector objective and the CSO penalty. This translates to setting $\lambda = 0.01$ for NC, $\lambda = 0.1$ for PT-RED, and $\lambda = 400$ for MMBD-CSO and MLBD-CSO.

**Evaluation Metrics.** We adopt detection accuracy to evaluate the performance for the considered detectors and also the effectiveness of the mixed label attack, compared with the other attacks. A successful detection requires both the backdoor attack to be detected and the target class to be correctly inferred. Clean models are used to evaluate the false positive detection rate.

## 3.2 Backdoor Detection Performance

For all datasets, we trained 10 clean models and 10 backdoored models with randomly chosen target classes and applied all detection methods for each model 5 times. (All the detectors involve random initialization for their optimization, so we repeated the experiment for 5 different seeds.) We calculate detection accuracy % (DA) over all 50 trials. The number of clean images per class is reported as $N_{img}$. Detection results where the CSO variant improves over its corresponding baseline by at least 20% are highlighted in gray, and the best performance for each attack type is shown in bold.

The results on all 3 data sets are shown in Tables 1-4. Surveying these tables, we make the following key observations: 1) CSO variants substantially outperform their baselines. For example, MMBD-CSO achieves much higher DA than MMBD in Table 2, *across all attacks*. Likewise, CSO variants of PT-RED and NC substantially outperform their baselines; 2) CSO variants have low false positives, particularly MMBD-CSO. It achieves just 4% in Tables 1 and 2, 14% in Table 3 and 12% in Table 4. By comparison, UNICORN and BTI-DBF give much higher false positives (even though BTI-DBF uses many more (e.g. 250 for CIFAR-10) clean images than the CSO methods); 3) MMBD-CSO is the overall best-performing detector, across these experiments, with both higher (in many cases much higher) DAs overall and lower false positives overall than all other detectors; 4) MLBD-CSO fares worse than MMBD-CSO, which suggests that margin, rather than logit, maximization yields a more robust detection statistic; 5) One-to-one and multi-to-one attacks are consistently harder to detect than all-to-one attacks at similar poisoning rates (PRs), as reflected by lower DA across datasets and detectors. Moreover, surveying Table 2, 3 and 4, ML attacks are much harder to detect than the baseline dirty-label attacks (even though CPR=DPR, *i.e.* the amount of clean label poisoning is very low). For example, in Table 2, with very few exceptions, ML detection accuracies are *substantially* lower than baselines, across all attack types, and against all detectors. This also holds true for the seven-to-one attacks in Tables 3 and 4. For example, MMBD-CSO's detection accuracy drops from 96% for 1-pixel to 68% for ML-1-pixel in Table 3. 6) For these low poisoning rate experiments, detection rates are much lower than reported in original papers, where higher poisoning rates were used. For example, UNICORN reported 95% detection accuracy for BadNet with 5% poisoning rate, but with 0.1% poisoning rate the detection accuracy is only 54%; BTI-DBF reported 100% detection accuracy with 5% poisoning rate, but with 0.1% poisoning rate the detection accuracy is only 72%. MMBD and MMBD-CSO both achieve much higher detection rates at a higher poisoning rate, as shown in Appendix A.7.

## 3.3 Adaptive Attack

We further examine whether our CSO defense can be compromised when the adversary has knowledge of the method. Class subspace feature decoupling and CSO leverage the assumption that backdoor and intrinsic features are separable in the latent space. To exploit this, we consider two types of adaptive attacks. First, following (Xu et al. (2024)), we adopt Adaptive-Blend (Qi et al., 2023), which diminishes the latent separation between clean and poisoned samples. Second, we introduce a new adaptive attack, Adaptive-Blend-2, designed to undermine CSO by choosing the backdoor trigger to contain intrinsic features of the target class. Specifically, for CIFAR-10, we consider 'dog' as the backdoor target class, with the backdoor pattern a $16 \times 16$ pixel dog's face. For each poisoned training image, we randomly crop the backdoor to a $8 \times 8$ patch and blend this patch into the source class image. By using a different random cropping for each poisoned image, we are capturing *all* of the dog's intrinsic features, across the collection of poisoned training images. At the same time, we are limiting the backdoor pattern to $8 \times 8$ ($16 \times 16$ would be too large, given the small size of

| CIFAR-10 | | All-to-one attack | | | | | | | | |
|---|---|---|---|---|---|---|---|---|---|---|
| Detector | $N_{\mathrm{img}}$ | clean | BadNet | chess | 1-pixel | blend | WaNet | IAD | LC | Bpp |
| UNICORN | 10 | 52 | 54 | 52 | 38 | 76 | 56 | 34 | 74 | **40** |
| BTI-DBF | 250 | 0 | 72 | **100** | **100** | **100** | 34 | 32 | 4 | 6 |
| BTI-DBF-2 | 250 | 40 | 12 | 60 | 76 | 48 | 20 | 32 | 0 | 0 |
| NC | 10 | 72 | 90 | **100** | **100** | 82 | 26 | 44 | 0 | 0 |
| NC-CSO | 10 | 70 | **100** | **100** | **100** | 100 | 54 | 74 | 4 | 18 |
| PT-RED | 100 | 62 | 0 | **100** | **100** | 90 | 46 | 4 | 10 | 10 |
| PT-RED-CSO | 100 | 72 | 54 | **100** | **100** | 100 | 62 | 12 | 10 | 10 |
| MMBD | 0 | 86 | 82 | **100** | **100** | 98 | **90** | 76 | 30 | 0 |
| MMBD-CSO | 10 | **96** | 92 | **100** | **100** | 100 | 88 | **84** | **76** | 36 |
| MLBD-CSO | 10 | **96** | 90 | **100** | **100** | 100 | 84 | 66 | 56 | 32 |

Table 1: Detection accuracies for CIFAR-10 based on 10 clean models (for false-positive performance) and 10 attacked models for 9 different all-to-one backdoors. Highlighted cells indicate $\geq 20\%$ relative improvement of CSO over its baseline. Bold indicates best detector for a given attack.

| CIFAR-10 | | One-to-one attack | | | | | | | | | | | |
|---|---|---|---|---|---|---|---|---|---|---|---|---|---|
| Detector | $N_{\mathrm{img}}$ | clean | BadNet | ML-BadNet | chess | ML-chess | 1-pixel | ML-1-pixel | blend | ML-blend | WaNet | IAD | Bpp |
| UNICORN | 10 | 52 | 80 | 56 | 28 | 22 | 80 | 42 | 28 | 22 | 18 | 28 | **54** |
| BTI-DBF | 250 | 0 | 84 | 60 | 86 | 52 | **100** | 84 | 74 | 74 | **24** | **60** | 46 |
| BTI-DBF-2 | 250 | 40 | 42 | 30 | 60 | 44 | 66 | 36 | 44 | 10 | 0 | 0 | 0 |
| NC | 10 | 72 | 94 | 48 | 70 | 42 | 78 | 10 | 60 | 48 | 0 | 28 | 0 |
| NC-CSO | 10 | 70 | 90 | 66 | 82 | 54 | 96 | 38 | 76 | 74 | 14 | 56 | 8 |
| PT-RED | 100 | 62 | 10 | 12 | **100** | 90 | **100** | 84 | 90 | 78 | 10 | 40 | 0 |
| PT-RED-CSO | 100 | 72 | 12 | 16 | **100** | **100** | **100** | 92 | **98** | **90** | 10 | **60** | 12 |
| MMBD | 0 | 86 | 66 | 40 | 32 | 0 | 68 | 20 | 52 | 12 | 0 | 10 | 0 |
| MMBD-CSO | 10 | **96** | **96** | **70** | 78 | 66 | **100** | 94 | 84 | 78 | 20 | 48 | 36 |
| MLBD-CSO | 10 | **96** | 88 | 46 | 78 | 46 | **100** | 46 | 78 | 72 | 18 | 48 | 30 |

Table 2: CIFAR-10 detection accuracy under 11 one-to-one backdoors. Dirty-label and mixed-label (ML) settings are reported.

| GTSRB | | All-to-one attack | | | | | | | | | Seven-to-one attack | | | | | | | |
|---|---|---|---|---|---|---|---|---|---|---|---|---|---|---|---|---|---|---|
| Detector | $N_{\mathrm{img}}$ | clean | BadNet | chess | 1-pixel | blend | WaNet | IAD | LC | Bpp | BadNet | ML-BadNet | chess | ML-chess | 1-pixel | ML-1-pixel | blend | ML-blend |
| UNICORN | 10 | 46 | 12 | 10 | 34 | 32 | 82 | **42** | 22 | 46 | **98** | 42 | 50 | 38 | 84 | 34 | 98 | 90 |
| BTI-DBF | 60 | 0 | 18 | 18 | 36 | 40 | 36 | 12 | 20 | 76 | 88 | **74** | 78 | 20 | 80 | 48 | **100** | 62 |
| BTI-DBF-2 | 60 | 0 | 18 | 18 | 34 | 38 | 36 | 12 | 0 | 0 | 76 | 70 | 60 | 20 | 78 | 52 | 50 | 40 |
| NC | 10 | 62 | 72 | 90 | 72 | 54 | 72 | 10 | 42 | **100** | 90 | 44 | 44 | 18 | 40 | 44 | 60 | 58 |
| NC-CSO | 10 | 74 | **82** | **100** | 82 | 90 | 78 | 10 | 44 | **100** | 92 | 60 | 64 | 30 | 72 | 50 | 82 | 66 |
| PT-RED | 10 | 42 | 44 | 78 | 60 | 70 | 80 | 0 | 52 | 92 | 22 | 18 | 34 | 20 | 36 | 40 | 56 | 30 |
| PT-RED-CSO | 10 | 66 | 42 | 90 | 88 | 88 | 92 | 18 | 56 | 86 | 36 | 20 | 54 | 40 | 62 | 54 | 66 | 56 |
| MMBD | 0 | 76 | 52 | 10 | 60 | 78 | 88 | 24 | **100** | **100** | 70 | 40 | 18 | 10 | 72 | 38 | 52 | 68 |
| MMBD-CSO | 10 | **86** | 78 | 48 | **100** | 98 | **100** | 36 | **100** | **100** | 92 | 58 | **80** | 44 | 96 | 68 | **100** | 92 |

Table 3: Detection accuracies for GTSRB on clean models and attacked models, for 9 all-to-one backdoors and 8 seven-to-one backdoors.

CIFAR-10 images). At test time, we evaluate the attack success rate by blending an $8 \times 8$ (dog) backdoor trigger pattern into test samples. We consider blending ratios from $0.2$ all the way up to $0.8$. In Table 5, we show the trigger-intrinsic overlap, which is the average rectified cosine similarity between the backdoored source class image features (masked to capture intrinsic features of the target class) and the target class clean image features. We also show the target class intrinsic feature overlap, which is the average rectified, masked feature cosine similarity between target class clean samples as a baseline for comparison. The trigger-intrinsic overlap becomes larger as the blend ratio increases.

| TinyImageNet | | All-to-one attack | | | | | | | | Seven-to-one attack | | | | | | | |
|---|---|---|---|---|---|---|---|---|---|---|---|---|---|---|---|---|---|
| Detector | $N_{\text{img}}$ | clean | BadNet | chess | 1-pixel | blend | WaNet | IAD | Bpp | BadNet | ML-BadNet | chess | ML-chess | 1-pixel | ML-1-pixel | blend | ML-blend |
| BTI-DBF | 25 | 68 | 24 | 86 | 90 | **100** | 12 | 16 | 4 | 28 | 40 | 86 | 28 | 82 | 50 | 94 | 66 |
| BTI-DBF-2 | 25 | 82 | 10 | 40 | 70 | 78 | 12 | 0 | 4 | 18 | 10 | 78 | 0 | 60 | 0 | 56 | 0 |
| NC | 10 | 10 | **100** | 74 | 76 | 86 | 38 | 0 | 30 | **100** | 84 | 96 | 52 | 44 | 30 | 50 | 52 |
| NC-CSO | 10 | 28 | **100** | 84 | 70 | **100** | 36 | 12 | 44 | **100** | **100** | **100** | 58 | 62 | 48 | 76 | 50 |
| PT-RED | 10 | **100** | 40 | 88 | 72 | 62 | 28 | 14 | 0 | 50 | 40 | **100** | 64 | 44 | 32 | 64 | 56 |
| PT-RED-CSO | 10 | **100** | 56 | **100** | **100** | **100** | 48 | 10 | 12 | 64 | 60 | **100** | **66** | 70 | 54 | 70 | 68 |
| MMBD | 0 | 78 | 96 | 28 | 94 | **100** | **100** | 0 | 62 | 96 | 60 | 18 | 0 | 60 | 30 | 50 | 46 |
| MMBD-CSO | 10 | 88 | 98 | 48 | **100** | **100** | **100** | **28** | **100** | **100** | 66 | 68 | 30 | **88** | **68** | **100** | **80** |

Table 4: Detection accuracies for TinyImageNet on clean models and attacked models, for 8 all-to-one backdoors and 8 seven-to-one backdoors.

| Blend ratio→ | 0.2 | 0.3 | 0.4 | 0.5 | 0.6 | 0.7 | 0.8 |
|---|---|---|---|---|---|---|---|
| Trigger-Intrinsic Feature Overlap | 0.40 | 0.40 | 0.54 | 0.52 | 0.60 | 0.53 | 0.64 |
| Target Class Intrinsic Feature Overlap | 0.67 | 0.63 | 0.65 | 0.65 | 0.68 | 0.64 | 0.67 |
| ASR(%) | 88.33 | 90.41 | 97.22 | 98.79 | 99.86 | 100.00 | 100.00 |
| ACC(%) | 91.20 | 91.35 | 91.34 | 91.51 | 91.23 | 91.44 | 91.50 |
| DA(%) | 100 | 100 | 100 | 100 | 96 | 96 | 90 |

Table 5: Results for Adaptive-Blend-2 with different blend ratios.

This is as one would expect. However, MMBD-CSO *still* achieves excellent detection accuracy, even up to a blend ratio of 0.8.

In Appendix A.3, Table 16 shows that MMBD-CSO also performs well against Adaptive-Blend, and we provide an illustration (Figure 3) for Adaptive-Blend-2.

### 3.4 ABLATION STUDIES

The defense considered here is MMBD-CSO, and the attacks are one-to-one BadNet and ML-BadNet. We use 10 clean and 10 backdoored ResNet-18 models on CIFAR-10 to assess detection results.

**Effects of Split Position for Feature Decoupling.** Table 6 shows the DA results of using different depths of the feature extractor $S_a$ – the 9th, 11th, 13th, 15th, and last convolution layers. As seen, splitting at later layers is generally better than splitting at earlier layers.

| Attack↓, layer→ | 9th | 11th | 13th | 15th | Last |
|---|---|---|---|---|---|
| clean | 88 | 90 | 96 | 98 | 96 |
| BadNet | 74 | 78 | 88 | 90 | 96 |
| ML-BadNet | 56 | 58 | 64 | 64 | 70 |

Table 6: DA for different split positions.

| Attack↓, $\lambda \to$ | 100 | 200 | 400 | 600 | 800 |
|---|---|---|---|---|---|
| clean | 84 | 90 | 96 | 96 | 90 |
| BadNet | 70 | 90 | 96 | 92 | 96 |
| ML-BadNet | 44 | 66 | 70 | 70 | 68 |

Table 7: DA for different CSO penalty values.

**Effects of CSO constraint penalty hyperparameter.** In Table 7, we evaluate the impact of $\lambda$. We observe that performance is not very sensitive, so long as $\lambda$ is not made too small. This is plausible because a larger $\lambda$ implies a greater degree of class subspace orthogonalization.

### 3.5 TIME COMPLEXITY

Figure 1 reports the execution time on CIFAR-10/ResNet-18 for the detectors compared in this work. The CSO variants introduce only a modest overhead compared to their baselines. The computing platform is specified in Appendix A.1.5.

### 3.6 ADDITIONAL EXPERIMENTAL RESULTS

Appendix A.2.1 shows that triggers are in general only weakly correlated with intrinsic features of the

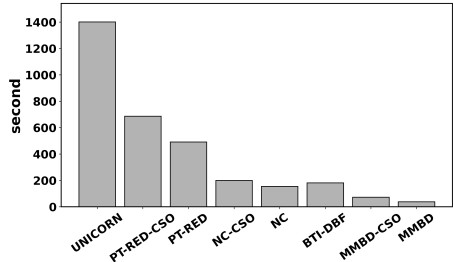

Figure 1: Execution time of the considered detectors.

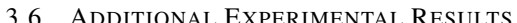

target class, which supports use of CSO. It also shows
how this correlation depends on the network layer. Appendix A.2.2 shows that there are in fact significant feature correlations between classes – in spite of this, CSO methods are quite successful at detection. Appendix A.4 includes additional ablation studies that show the effect of $N_{\mathrm{img}}$, the need for class-specific feature masking, and for applying the ReLU to the cosine similarity. In Appendix A.5, we evaluate MMBD-CSO against additional stealthy attacks, clean label attacks, and multi-trigger attacks. In A.7 we evaluate MMBD-CSO against WaNet, IAD, and LC at a higher poisoning rate. These results (with high detection rates) show that the low detection rates for these attacks in Tables 1 and 2 are largely due to the use of an unusually low poisoning rate. We also evaluate MMBD-CSO against larger DNN architectures VGG-16 and ViT in Appendix A.6. Moreover, we evaluate MMBD-CSO when the clean set used for detection is subject to domain shift or to mislabeling (Appendix A.8). Finally, in Appendix A.9, we discuss in detail how our mixed-label attack affects collateral damage and detection performance.

## 4 RELATED WORK

### 4.1 BACKDOOR DEFENSE

Feature masking in backdoor detection has been widely applied, as early as (Liu et al. (2018); Wang et al. (2019)), and more recently in (Huang et al. (2022); Xu et al. (2024); Wang et al. (2022a; 2023)), and our use of it is inspired by these earlier works. However, unlike (Xu et al. (2024); Wang et al. (2022a; 2023)), our approach learns a *class-dependent* soft feature mask, in order to identify a putative target class's intrinsic, discriminating features. Note also that our method, using class-dependent masks, outperforms (Huang et al. (2022); Xu et al. (2024); Wang et al. (2022a; 2023)), which use global masks. The necessity of using a class-dependent mask for CSO is demonstrated in Appendix A.4.

We first came up with the CSO approach by considering detectors such as MMBD (Wang et al. (2024)), which have no mechanism for exploiting an available, small clean set of samples. We sought to redress that limitation. While CSO gives such a mechanism, we further recognized its wider applicability, beyond just improving MMBD. While cosine similarity has been part of a backdoor defense previously (Zeng et al. (2024)), its use there was to effectively encourage clustering in activation space – quite different from class subspace orthogonalization.

### 4.2 BACKDOOR ATTACKS

Clean-label attacks are stealthier than dirty-label attacks in that they do not require any mislabeling. This may be accompanied by strategies encouraging the model to associate the trigger with the target class (Turner et al. (2018)), (Zhao et al. (2020)),Souri et al. (2022)). We use clean-label poisoning for a different purpose – not to make source classes trigger-susceptible, but to eliminate "collateral damage". Moreover, as shown here, even with very modest CPR, this attack is much harder to detect than a pure dirty-label attack. Also, unlike adaptive dirty-label attacks or sophisticated clean-label strategies (Turner et al. (2018)), (Zhao et al. (2020)),Souri et al. (2022)), our attack is *passive*: it requires only a poisoning capability, without assuming the attacker is the training authority, has surrogate model access, or access to a large amount of data from the domain.

## 5 CONCLUSIONS

In this work, we developed a general "plug-and-play" framework – class subspace orthogonalization (CSO) – for enhancing the sensitivity of backdoor detectors. We evaluated CSO-enhanced detectors and baselines against a variety of existing attacks, as well as against a novel mixed dirty/clean label attack, proposed here, that is significantly harder to detect than traditional dirty-label attacks. Evaluation on the mixed label attack shows that existing detectors are inadequate, and CSO is necessary, to help detect subtle, stealthy threats. In future, we may consider whether CSO can also benefit existing backdoor *mitigation* techniques.

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

# A  APPENDIX

## A.1  MORE DETAILS OF EXPERIMENT SETTINGS

### A.1.1  DATASETS

**CIFAR10 (Krizhevsky & Hinton (2009)).** This dataset is built for recognizing general objects such as cats, deer, and automobiles. It contains 50,000 training samples and 10,000 test samples in 10 classes.

**GTSRB (Houben et al. (2013)).** The German Traffic Sign Recognition Benchmark is designed for traffic sign classification. It contains 39,209 training images and 12,630 test images across 43 classes of traffic signs.

**TinyImageNet (Le & Yang (2015)).** This dataset is a scaled-down version of the ImageNet (Russakovsky et al. (2015)) benchmark. It consists of 100,000 training samples and 10,000 validation samples over 200 object categories, each with 500 training images and 50 validation images. In this paper, we select a subset of the original TinyImageNet which contains the first 40 classes.

### A.1.2  SETTINGS FOR DETECTION METHODS

In this section, we describe the detailed settings of the detectors used in our experiments.

**UNICORN (Wang et al. (2023)).** We follow the default settings used in its original paper. Also, we determine a model is clean, i.e., without a backdoor, if the ASR-Inv of all labels is no larger than 90%.

**BTI-DBF (Xu et al. (2024)).** We follow the default settings used in its original paper. BTI-DBF *always* detects a backdoor is present since it predicts the target label by selecting the class with the highest frequency over the model's predictions. In order to endow detection specificity to BTI-DBF, we define a threshold for this frequency ratio. Specifically, for BTI-DBF results reported in our experiments, if the maximum frequency ratio exceeds $1/|\mathcal{K}|$ (the uniform random baseline over $|\mathcal{K}|$ classes), it is considered an indicator of a potential backdoor. In BTI-DBF-2, we instead set this threshold to $1.0$. The former choice prioritizes backdoor detection sensitivity, while the latter aims to minimize false positives in clean models.

**NC (Wang et al. (2019)).** We follow the default settings used in its original paper.

**NC-CSO.** We set $\lambda = 0.01$ in Eq. 8. All remaining settings are kept consistent with NC (Wang et al. (2019)).

**PT-RED (Xiang et al. (2022)).** We follow the default settings used in its original paper.

**PT-RED-CSO.** We set $\lambda = 0.1$ in Eq. 9. All remaining settings are kept consistent with PT-RED (Xiang et al. (2022)).

**MMBD (Wang et al. (2024)).** We follow the default settings used in its original paper.

**MMBD-CSO.** We set $\lambda = 400$ in Eq. 6. All remaining settings are kept consistent with MMBD (Wang et al. (2024)).

**MLBD-CSO.** We set $\lambda = 400$ in Eq. 7. All remaining settings are kept consistent with MMBD (Wang et al. (2024)), except that we replace the maximum margin objective with the maximum logit objective.

### A.1.3  SETTINGS FOR BACKDOOR ATTACKS

In this section, we detail the backdoor attacks considered in our experiments, including the poisoning rate, clean test accuracy, and, for one-to-one and multi-to-one attacks, collateral damage.

**BadNets (Gu et al. (2019)).** We consider a $3 \times 3$ random patch for CIFAR-10 and GTSRB and a $5 \times 5$ random patch for TinyImageNet, with a randomly selected location, which is fixed for all trigger images for a given attack.

**Chessboard (Xiang et al. (2022)).** We consider a global additive perturbation (with size 3/255) resembling a chessboard.

**1-pixel (Tran et al. (2018)).** For CIFAR-10 and GTSRB, we consider a 1-pixel additive backdoor pattern which perturbs a single, randomly selected pixel by 75/255 in all color channels, with this pixel location fixed for all trigger images for a given attack. For TinyImageNet, we consider a 4-pixel pattern that perturbs 4 randomly selected pixels.

**Blend (Wang et al. (2019)).** We consider a $3 \times 3$ local random patch trigger with a blend ratio of $\alpha = 0.2$, with a randomly selected and fixed location for each image in a given attack.

**WaNet (Nguyen & Tran (2021)).** We consider a warping-based trigger with the default settings in the original paper.

**IAD (Nguyen & Tran (2020)).** We consider an input-aware dynamic trigger that perturbs each image using a small, input-dependent noise pattern, following the settings in the original paper.

**LC (Turner et al. (2019)).** We use projected gradient descent (PGD) to make adversarial samples and set the maximum perturbation size $\epsilon = 8$.

**Bpp (Wang et al. (2022b)).** We consider a trigger that utilizes image quantization and dithering techniques with the default settings in the original paper.

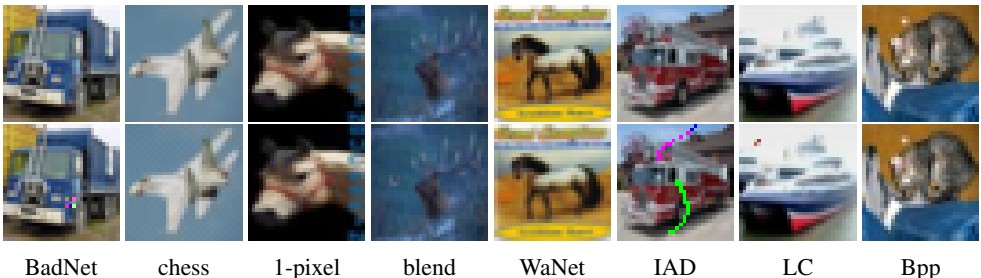

| BadNet | chess | 1-pixel | blend | WaNet | IAD | LC | Bpp |

Figure 2: Example benign images (top) and corresponding poisoned images (bottom) used in our experiments.

In Figure 2, we demonstrate examples of poisoned images generated by different attacks on CIFAR-10.

Tables 8, 9, 10, 11, 12 and 13 give the poisoning rate (PR) for each attack (or for ML backdoors the dirty-label poisoning rate (DPR)), the average attack success rate (ASR) as well as the clean test accuracy (ACC) over the 10 clean models and 10 attacked models. *We set the poisoning rate to the minimum required to achieve a high attack success rate ($\approx 90\%$).* This choice makes the attack substantially harder to detect than the attacks (with much higher poisoning rates) used in the original defense papers, which explains the observed gap in detection accuracy.

For ML backdoors, the clean poisoning rate (CPR) equals the DPR. For one-to-one attacks on CIFAR-10 and seven-to-one attacks on GTSRB and TinyImageNet, we also show the average collateral damage (CD), which is calculated as the ratio of the non-source (and non-target) class samples, with trigger embedded, that are misclassified to the target class divided by the number of non-source (and non-target) class samples. Note that ML backdoor attacks have much lower CD than the non-ML backdoors.

Note also that DPR here is defined as the ratio of the number of dirty-label poisoned training samples from the source class to the total number of training samples. This definition ensures compatibility with PR, maintaining consistency in how poisoning levels are measured across different attack setups.

### A.1.4 SETTINGS FOR MODEL TRAINING

For BadNet, chessboard, 1-pixel and blend attacks, we adopt ResNet-18 (He et al. (2016)) for CIFAR-10, MobileNet (Howard et al. (2017)) for GTSRB, and ResNet-34 (He et al. (2016)) for TinyImageNet. For LC, we adopt ResNet-18 for CIFAR-10 and PreActResNet-18 (Yu et al. (2018)) for GTSRB and TinyImageNet. For WaNet, IAD and Bpp, we adopt PreActResNet-18 for all datasets.

| CIFAR-10 | | | | | | | | | |
|---|---|---|---|---|---|---|---|---|---|
| All-to-one | clean | BadNet | chess | 1-pixel | blend | WaNet | IAD | LC | Bpp |
| PR(%) | 0 | 0.1 | 0.5 | 0.5 | 5.0 | 5.0 | 5.0 | 5.0 | 1.0 |
| ASR(%) | 0 | 99.92 | 99.94 | 91.28 | 96.86 | 84.46 | 91.68 | 88.52 | 99.58 |
| ACC(%) | 91.61 | 91.59 | 91.06 | 91.12 | 91.48 | 90.72 | 91.23 | 91.77 | 91.00 |

Table 8: On CIFAR-10, the poisoning rate (PR), average attack success rate (ASR) and average clean test accuracy (ACC) over the ensemble of 10 clean models and 10 attacked models for all-to-one backdoors.

| CIFAR-10 | | | | | | | | | | | |
|---|---|---|---|---|---|---|---|---|---|---|---|
| One-to-one | clean | BadNet | ML-BadNet | chess | ML-chess | 1-pixel | ML-1pixel | blend | ML-blend | WaNet | IAD | Bpp |
| PR/DPR(%) | 0 | 0.1 | 0.1 | 0.2 | 0.2 | 3.0 | 3.0 | 2.0 | 2.0 | 5.0 | 3.0 | 1.0 |
| ASR(%) | 0 | 92.16 | 92.11 | 90.53 | 93.98 | 93.01 | 92.01 | 90.52 | 91.67 | 90.96 | 87.04 | 100.00 |
| CD(%) | 0 | 54.27 | 24.55 | 58.36 | 24.44 | 36.38 | 4.26 | 41.75 | 6.48 | 3.64 | 25.99 | 2.52 |
| ACC(%) | 91.61 | 91.84 | 90.85 | 91.43 | 90.22 | 92.68 | 91.63 | 91.32 | 90.47 | 90.60 | 90.41 | 91.61 |

Table 9: On CIFAR-10, PR (or for ML backdoors the dirty-label poisoning rate (DPR)), ASR, average collateral damage (CD) and ACC over the ensemble of 10 clean models and 10 attacked models for one-to-one backdoors.

| GTSRB | | | | | | | | | |
|---|---|---|---|---|---|---|---|---|---|
| All-to-one | clean | BadNet | chess | 1-pixel | blend | WaNet | IAD | LC | Bpp |
| PR(%) | 0 | 0.1 | 1.1 | 1.1 | 0.5 | 5.0 | 5.0 | 3.8 | 1.0 |
| ASR(%) | 0 | 97.90 | 97.32 | 90.76 | 98.09 | 98.18 | 98.26 | 86.63 | 81.21 |
| ACC(%) | 95.27 | 95.06 | 95.04 | 94.91 | 95.16 | 95.09 | 95.36 | 84.26 | 95.94 |

Table 10: On GTSRB, PR, ASR and ACC over the ensemble of 10 clean models and 10 attacked models for all-to-one backdoors.

| GTSRB | | | | | | | | | |
|---|---|---|---|---|---|---|---|---|---|
| Seven-to-one | clean | BadNet | ML-BadNet | chess | ML-chess | 1-pixel | ML-1pixel | blend | ML-blend |
| PR/DPR(%) | 0 | 3.2 | 3.2 | 8.1 | 8.1 | 3.2 | 3.2 | 3.2 | 3.2 |
| ASR(%) | 0 | 100.00 | 99.14 | 96.80 | 96.88 | 96.99 | 96.43 | 93.77 | 95.77 |
| CD(%) | 0 | 98.84 | 6.54 | 16.33 | 1.04 | 85.75 | 4.09 | 10.88 | 1.59 |
| ACC(%) | 95.27 | 95.58 | 93.12 | 94.93 | 94.26 | 95.27 | 96.43 | 94.64 | 94.52 |

Table 11: On GTSRB, PR/DPR, ASR, CD and ACC over the ensemble of 10 clean models and 10 attacked models for seven-to-one backdoors.

| TinyImageNet | | | | | | | | |
|---|---|---|---|---|---|---|---|---|
| All-to-one | clean | BadNet | chess | 1-pixel | blend | WaNet | IAD | Bpp |
| PR(%) | 0 | 1.0 | 3.0 | 5.0 | 10.0 | 5.0 | 5.0 | 1.0 |
| ASR(%) | 0 | 95.69 | 90.83 | 89.08 | 95.17 | 98.57 | 91.33 | 89.1 |
| ACC(%) | 66.46 | 66.59 | 65.81 | 65.53 | 65.05 | 63.88 | 63.79 | 62.77 |

Table 12: On TinyImageNet, PR, ASR and ACC over the ensemble of 10 clean models and 10 attacked models for all-to-one backdoors.

We train all DNNs with the Adam optimizer, using a learning rate of 0.001 and a batch size of 128, for 100 epochs on each dataset.

### A.1.5 COMPUTING PLATFORM

In this paper, all experiments are conducted on a single NVIDIA RTX 3090 Ti GPU using PyTorch.

| TinyImageNet | | | | | | | | | |
|---|---|---|---|---|---|---|---|---|---|
| Seven-to-one | clean | BadNet | ML-BadNet | chess | ML-chess | 1-pixel | ML-1pixel | blend | ML-blend |
| PR/DPR(%) | 0 | 1.8 | 1.8 | 5.3 | 5.3 | 5.3 | 5.3 | 14.0 | 14.0 |
| ASR(%) | 0 | 98.61 | 85.21 | 97.77 | 84.74 | 93.60 | 84.69 | 90.52 | 86.05 |
| CD(%) | 0 | 90.43 | 34.14 | 91.22 | 25.00 | 82.75 | 28.95 | 83.28 | 20.12 |
| ACC(%) | 66.46 | 65.37 | 64.70 | 64.52 | 64.83 | 64.20 | 64.94 | 65.98 | 66.07 |

Table 13: On TinyImageNet, PR/DPR, ASR, CD and ACC over the ensemble of 10 clean models and 10 attacked models for seven-to-one backdoors.

## A.2 EMPIRICAL ANALYSIS OF FEATURE OVERLAP

In this section, we provide an empirical analysis of feature overlap in backdoored models. We examine two complementary aspects: (i) trigger–intrinsic overlap, which measures how similar backdoor-induced features are to the intrinsic features of the target class across different network layers, and (ii) cross-class intrinsic overlap, which quantifies the inherent feature similarity between source classes and the target class. Together, these analyses help assess the separability of backdoor features from intrinsic features of a target class and provide empirical support for the design assumptions underlying CSO.

### A.2.1 QUANTITATIVE RESULTS ON LAYERWISE TRIGGER-INTRINSIC FEATURE OVERLAP BETWEEN BACKDOOR FEATURES AND TARGET CLASS INTRINSIC FEATURES

In this section, we quantify "trigger–intrinsic" overlap, i.e., the separability of trigger features from intrinsic target class features, and investigate the overlapping layer-wise. To be specific, the overlap can be experimentally assessed by measuring the average rectified, masked feature cosine similarity, in various layers, between samples from non-target classes that contain the backdoor trigger and samples from the target class. We refer to this as the trigger-intrinsic feature overlap. Table 14 shows that the correlations between samples (without the trigger) from the target class – target class intrinsic feature overlap – are larger than the correlations between source class samples with the trigger and target class samples (without the trigger). This is as one would expect, and is supportive of CSO's key assumption – that the intrinsic feature overlap between source class samples with the trigger and target class samples is low. In terms of the layer, we can see that, for most of the attacks (excepting IAD and Bpp), the trend is that the correlations in deeper layers ($13^{th}$ and $17^{th}$) are smaller than in the earlier layers. This gives some empirical support to our choice of a deep layer for measuring the CSO penalty.

### A.2.2 QUANTITATIVE RESULTS ON CROSS-CLASS FEATURE OVERLAP BETWEEN SOURCE CLASS FEATURES AND TARGET CLASS INTRINSIC FEATURES

In this section, we discuss intrinsic feature overlap across classes. This overlap can be experimentally understood by measuring, for different attacks, average rectified, masked feature cosine similarity between samples from source classes and samples from the target class. We show these results for CIFAR-10 in Table 15. We also evaluate the target class intrinsic feature overlap, i.e., average rectified, masked feature cosine similarity between target class samples as a baseline for comparison. The results show that, while the source-target correlations are smaller than the target-target correlations, there are significant source-target correlations under different attacks. This is in fact not so surprising, since different CIFAR-10 classes share high-level attributes. For example, dogs, cats, horses, and frogs all have eyes, and they all have legs. Despite these "feature overlaps", MMBD-CSO is achieving strong detection results, and much better than existing baselines, against an array of attacks on this (CIFAR-10) data set.

## A.3 MORE DETAILS ON ADAPTIVE ATTACKS

In this section, we provide more details on the adaptive attacks considered in Section 3.4. We evaluate two attacks on CIFAR-10, **Adaptive-Blend** and **Adaptive-Blend-2**, both aiming to undermine the assumption that class-specific feature decoupling can cleanly separate backdoor and intrinsic features.

| Attack | Layer | Trigger-Intrinsic Feature Overlap | Target Class Intrinsic Feature Overlap |
|---|---|---|---|
| BadNet | 5th | 0.38 | 0.46 |
| | 9th | 0.26 | 0.32 |
| | 13th | 0.21 | 0.33 |
| | 17th | 0.32 | 0.56 |
| chess | 5th | 0.40 | 0.40 |
| | 9th | 0.30 | 0.32 |
| | 13th | 0.21 | 0.31 |
| | 17th | 0.29 | 0.50 |
| 1-pixel | 5th | 0.40 | 0.42 |
| | 9th | 0.31 | 0.36 |
| | 13th | 0.28 | 0.39 |
| | 17th | 0.27 | 0.71 |
| blend | 5th | 0.39 | 0.45 |
| | 9th | 0.26 | 0.32 |
| | 13th | 0.22 | 0.34 |
| | 17th | 0.21 | 0.52 |
| WaNet | 5th | 0.40 | 0.44 |
| | 9th | 0.48 | 0.53 |
| | 13th | 0.53 | 0.56 |
| | 17th | 0.33 | 0.61 |
| IAD | 5th | 0.23 | 0.26 |
| | 9th | 0.49 | 0.52 |
| | 13th | 0.53 | 0.59 |
| | 17th | 0.58 | 0.74 |
| LC | 5th | 0.40 | 0.51 |
| | 9th | 0.26 | 0.33 |
| | 13th | 0.17 | 0.28 |
| | 17th | 0.22 | 0.77 |
| Bpp | 5th | 0.17 | 0.28 |
| | 9th | 0.53 | 0.55 |
| | 13th | 0.46 | 0.58 |
| | 17th | 0.63 | 0.73 |

Table 14: Layerwise trigger–intrinsic feature overlap and target-class intrinsic feature overlap for different backdoor attacks.

| Attack→ | BadNet | chess | 1-pixel | blend | WaNet | IAD | LC | Bpp |
|---|---|---|---|---|---|---|---|---|
| Source Class, Target Class Intrinsic Feature Overlap | 0.39 | 0.31 | 0.29 | 0.28 | 0.34 | 0.24 | 0.29 | 0.26 |
| Target Class Intrinsic Feature Overlap | 0.56 | 0.50 | 0.71 | 0.52 | 0.61 | 0.74 | 0.77 | 0.73 |

Table 15: Source class feature and target class intrinsic feature overlap, and target class intrinsic feature overlap for different backdoor attacks.

**Adaptive-Blend (Qi et al. (2023)).** This attack generates partial-blend patterns as triggers for model training, and additionally includes regularized training samples in which backdoor patterns are embedded but labeled correctly, thereby circumventing defenses that rely on latent separability. We followed the original settings in the paper and generated 10 models. As seen in Table 16, our defense performs very well against Adaptive-Blend.

| Attack | ACC(%) | ASR(%) | DA(%) |
|---|---|---|---|
| Adaptive-Blend | 91.73 | 90.56 | 98 |

Table 16: Results on Adaptive-Blend.

**Adaptive-Blend-2.** As described in Section 3.4, since our CSO variants rely on isolating backdoor information from target-class intrinsic features, we design an attack that explicitly leverages intrinsic content. Figure 3 shows an illustration of Adaptive-Blend-2.

Figure 3: Illustration of our adaptive attack Adaptive-Blend-2. The target class is 'dog' and the backdoor trigger is a $16 \times 16$ dog's face. To poison the training set, we generate sample-specific triggers by randomly cropping $8 \times 8$ regions from the original $16 \times 16$ dog-face pattern and blending them into source-class images. We considered attacks with blend ratios from $0.2$ all the way up to $0.8$. The figure is showing the blending into two different source class (cat) images. Note, particularly in the upper row, that the dog features become quite visible as the blend ratio is increased, i.e. the attack is becoming less "stealthy".

As shown in Table 5, CSO remains highly effective under Adaptive-Blend-2 even up to a blend ratio of 0.8, highlighting its robustness against adaptive strategies.

### A.4 RESULTS OF ADDITIONAL ABLATION STUDIES

In this section, we present additional ablation studies for MMBD-CSO on CIFAR-10 under all-to-one attacks.

**Effect of $N_{\text{img}}$.** In Table 17, we show the detection results when the number of available clean images is varied. Good results are achieved even with $N_{\text{img}} = 5$.

**Effect of Class-specific decoupling.** In Table 18, we demonstrate that class-specific feature masking is essential for CSO to achieve effective detection. When a single global mask is estimated across all classes, detection performance drops for both BadNet and, especially, for ML-BadNet.

**Effect of ReLU function in Eq. 5.** In Table 19, we show the necessity of the ReLU function acting on the cosine similarity.

| Attack↓, $N_{img}$→ | 1 | 5 | 10 | 50 | 100 |
|---|---|---|---|---|---|
| clean | 70 | 92 | 96 | 94 | 100 |
| BadNet | 84 | 96 | 96 | 98 | 96 |
| ML-BadNet | 44 | 68 | 70 | 66 | 72 |

Table 17: Detection accuracy for different number of clean samples.

| Attack | non-class-specific | class-specific |
|---|---|---|
| clean | 98 | 96 |
| BadNet | 88 | 96 |
| ML-BadNet | 36 | 70 |

Table 18: Detection accuracy with non-class-specific and class-specific feature masking.

| Attack | w/o ReLU | w/ ReLU |
|--------|----------|---------|
| clean | 96 | 96 |
| BadNet | 70 | 96 |
| ML-BadNet | 32 | 70 |

Table 19: Detection accuracy without or with the ReLU function applied to the CSO penalty.

### A.5 ADDITIONAL EXPERIMENTAL RESULTS AGAINST MORE ATTACKS

In this section, we present additional experimental results against other stealthy attacks, clean label attacks, and multi-trigger attacks for MMBD and MMBD-CSO on CIFAR-10. All attacks considered are all-to-one. We ran MMBD and MMBD-CSO on each model 5 times.

#### A.5.1 RESULTS ON DRUPE ATTACK (TAO ET AL. (2024))

The DRUPE (Tao et al. (2024)) attack is designed to make out-of-distribution, highly clustered poisoned samples blend into the clean data distribution and also to be more dispersed.

**Settings.** We reproduced the DRUPE attack with its official code under the default settings, where the encoder is pretrained on CIFAR-10 and downstream classifiers are trained on GTSRB. We generated five backdoored downstream models with 2 different reference inputs for each model.

**Results.** MMBD achieved a detection accuracy of only 4%, while MMBD-CSO achieved 60%, demonstrating a huge improvement over the baseline MMBD method for this challenging, advanced attack.

#### A.5.2 RESULTS ON BYPASSING ATTACK(SHOKRI ET AL. (2020))

The Bypassing Attack (Shokri et al. (2020)) is designed to maximize the indistinguishability of the latent representations of poisoned data and clean data.

**Settings.** We implemented the Bypassing attack (Shokri et al. (2020)) on ten BadNet baseline models with randomly chosen target labels and then jointly trained the backdoored model with the discriminator network to suppress feature differences between backdoored and clean samples, following the experimental configuration described in the original paper.

**Results.** MMBD detected no backdoors (DA = 0%), while MMBD-CSO achieved detection accuracy of 40%, demonstrating a huge improvement over the baseline MMBD method on this challenging attack.

#### A.5.3 RESULTS ON MORE CLEAN-LABEL ATTACKS

In this section, we evaluate our methods on more types of clean label attacks. To be specific, we consider Refool (Liu et al. (2020)), Narcissus (Zeng et al. (2023)), and SIG (Barni et al. (2019)) attacks. Refool creates triggers by simulating realistic environment reflections on objects. Narcissus implants a backdoor signature directly into the model's representation space by learning a perturbation for some training samples so that their hidden-layer activations become closer to a specific target signature. SIG generates triggers using a global sinusoidal perturbation.

**Settings.** We follow the default settings used in the original papers. 10 models for each attack are generated with randomly chosen target class.

**Results.** In Table 20, we show that MMBD-CSO gives DA improvement over MMBD across these attacks.

#### A.5.4 RESULTS ON MULTI-TRIGGER ATTACK(LI ET AL. (2024))

(Li et al. (2024)) propose a multi-trigger attack MTBA where multiple adversaries use different types of triggers to poison the same dataset.

| CIFAR-10 | Refool | Narcissus | SIG |
|----------|--------|-----------|-----|
| MMBD | 76 | 16 | 100 |
| MMBD-CSO | 94 | 56 | 100 |

Table 20: Detection accuracy for more clean label backdoor attacks.

**Settings.** We considered all-to-one MTBA with 10 different triggers and followed the default settings used in the original paper. 10 backdoored models are trained with a randomly chosen target class.

**Results.** MMBD achieved 90% detection accuracy, while MMBD-CSO achieved 98%, which demonstrates the CSO method's resistance to multi-trigger attacks.

### A.6 ADDITIONAL EXPERIMENTAL RESULTS WITH MORE MODEL ARCHITECTURES

In this section, we present additional experiments using more representative model architectures. Specifically, we evaluate our methods on CIFAR-10 using VGG-16 (Simonyan & Zisserman (2015)) and ViT (Dosovitskiy et al. (2021)) (without pre-training) under both BadNet and ML-BadNet attacks. As shown in Table 21, MMBD-CSO achieves substantially higher detection accuracy compared with MMBD. These results further demonstrate the generalizability and robustness of our proposed approach across diverse neural network architectures.

| Model | Attack | MMBD | MMBD-CSO |
|-------|--------|------|----------|
| VGG-16 | BadNet | 68 | 92 |
| | ML-BadNet | 0 | 36 |
| ViT | BadNet | 42 | 78 |
| | ML-BadNet | 0 | 20 |

Table 21: Detection accuracy on larger model architectures.

### A.7 RESULTS ON WANET, IAD, AND LC WITH HIGHER POISONING RATE

The detection rates in the main paper for all methods are all "unusually" low because we chose a low poisoning rate (to make the detection problem as challenging as possible). When higher poisoning rates are chosen, all methods generally achieve higher true detection rates. Much higher poisoning rates were used in the original papers for the baseline methods, and much higher detection rates were reported in these original papers. We now demonstrate this for MMBD and MMBD-CSO in Table 22. Here, the poisoning rate was increased to 10 % and the detection rates are much higher on WaNet, IAD, LC, and Bpp than the results reported in Table 1. At the same time, MMBD-CSO is still achieving a detection advantage over MMBD – at this higher poisoning rate, it is achieving excellent performance in detecting these stealthy attacks.

| CIFAR-10 | WaNet | IAD | LC | Bpp |
|----------|-------|-----|-----|-----|
| MMBD | 98 | 88 | 78 | 66 |
| MMBD-CSO | 96 | 96 | 100 | 88 |

Table 22: Detection accuracy for all-to-one WaNet, IAD, LC and Bpp attacks with higher poisoning rates of 10%.

### A.8 EFFECTS OF LABEL NOISE AND DOMAIN SHIFT ON THE SMALL CLEAN SET

**Effect of label noise.** In Table 23, we consider mislabeling up to 30% of the clean set samples used by MMBD-CSO. The results show that MMBD-CSO is robust against such "label noise".

| Mislabeling Fraction | 0% | 10% | 20% | 30% |
|---|---|---|---|---|
| BadNet | 96 | 96 | 90 | 90 |
| ML-BadNet | 70 | 72 | 68 | 66 |

Table 23: Detection accuracy on clean data with different mislabeling fractions.

**Effect of domain shift.** In Table 24, we conduct two types of domain shift wherein the clean samples are altered to introduce either color jitter or Gaussian blurring. The results show that our method is fully robust to both types of domain shift.

| Domain shift | w/o | Color jitter | Guassian blur |
|---|---|---|---|
| BadNet | 96 | 94 | 96 |
| ML-BadNet | 70 | 70 | 74 |

Table 24: Detection accuracy on clean data with domain shift.

## A.9   VARYING THE CPR FOR MIXED LABEL ATTACKS

Our mixed-label attack is designed to reduce unintended misclassification of trigger-bearing, non-source class samples to the target class. Below we assess how varying the clean-label poisoning rate (CPR) under a fixed dirty-label poisoning rate (DPR) impacts collateral damage and detection accuracy. For clarity and brevity we focus on ML-BadNet here; experiments with other mixed-label attacks show comparable trends.

As shown in Table 25, mixed-label BadNet significantly reduces collateral damage compared to the purely dirty-label BadNet, with collateral damage decreasing further as CPR increases. Notably, as CPR is varied, ML-BadNet preserves high backdoor attack success rates while leaving clean accuracy largely unaffected. In contrast, MMBD-CSO exhibits a clear drop in detection accuracy between BadNet and ML-BadNet, and the gap widens as CPR grows. These results reinforce that mixed-label attacks are substantially more difficult to detect than pure dirty-label attacks.

| | DPR = 0.1% | | | | |
|---|---|---|---|---|---|
| CPR(%) | 0 | 0.05 | 0.10 | 0.20 | 0.30 |
| ASR(%) | 92.16 | 93.61 | 92.11 | 87.14 | 88.85 |
| CD(%) | 54.27 | 29.343 | 24.55 | 12.46 | 11.77 |
| ACC(%) | 91.84 | 91.07 | 90.85 | 90.62 | 90.89 |
| DA(%) | 96 | 76 | 70 | 62 | 46 |

Table 25: Average ASR, CD, ACC and DA for different CPR under DPR $= 0.1\%$.

