# OpenReview forum: "Improving the Sensitivity of Backdoor Detectors via Class Subspace Orthogonalization"
_ICLR.cc/2026/Conference — Submitted to ICLR 2026_

### Official Review · Reviewer_tvn1 · 2025-10-28

**Soundness:** 3
**Presentation:** 3
**Contribution:** 2
**Rating:** 4
**Confidence:** 4

**Summary:**

This paper proposes a method called class subspace orthogonalization (CSO) to improve post-training backdoor detection by suppressing intrinsic class features for optimizing detector towards on backdoor-related directions. The authors argue that existing backdoor detection methods rate the success by an outlier statistic per class which fails when some non-target classes naturally yield extreme statistics or when the backdoor signal is subtle. The paper also gives a linear analysis showing orthogonalized optimization zeroes out non-target classes while preserving a positive statistic for the target class. They propose to estimate each class’s intrinsic feature subspace from a small clean set, then penalize alignment with that subspace during the detector’s search. This makes the target class stand out because only it contains both intrinsic and trigger contributions. The proposed method is implemented by learning class-specific soft masks to capture intrinsic features, adding a rectified cosine-similarity penalty to existing detectors to obtain the corresponding CSO variants. Experiments on standard datasets CIFAR-10, GTSRB, and TinyImageNet across multiple attacks show consistent gains and lower false positives with low overhead. The authors claim that the proposed method can substantially boost backdoor detector sensitivity and robustness against adaptive attackers.

**Strengths:**

Motivations are clear, the authors provide good explanations of why the target class remains an outlier after removing intrinsic features. The proposed method is easily integrated with existing backdoor detectors by adding a single penalty term. Experiments show good backdoor detection results with low overhead.

**Weaknesses:**

The approach assumes access to a small but representative clean set per class, and may be sensitivity to domain shift, label noise. As the motivation is based on backdoor features lying outside the intrinsic subspace, adaptive attacks that force overlap can escape it. Experiments use only moderate-scale backbones, it would be desirable to see results on larger modern models and vision transformers. The hyperparameter lambda are chosen per detector, it is not clear how to tune this parameter.

**Questions:**

1. How do CSO variant backdoor detectors depend on the number of images in the clean dataset per class? It would be helpful to better understand the relation between the choice of clean dataset and detector's performance.
2. If an adaptive attacker explicitly aligns trigger features with the target class’s intrinsic subspace, how can CSO be adapted?
3. It is shown that sensitivity to lambda is modest, but values differ by detector. Is there a way to efficiently tune this hyperparameter for different datasets and detectors?

---

> ### Author Response · Authors · 2025-11-23
> **Author Response to Reviewer tvn1 (Part 1 of 2)**
>
> Thanks for your time and reviewing effort.
> ___
> **Summary:** This paper proposes a method called class subspace orthogonalization (CSO) to improve post-training backdoor detection by suppressing intrinsic class features for optimizing detector towards on backdoor-related directions...Experiments on multiple datasets across multiple attacks show consistent gains and lower false positives with low overhead. The authors claim that the proposed method can substantially boost backdoor detector sensitivity and robustness against adaptive attackers.
>
> **Response:** Thank you for this accurate summary of our work.
>
>
> **Strengths:**
>
> **S1:** Motivations are clear, the authors provide good explanations of why the target class remains an outlier after removing intrinsic features. The proposed method is easily integrated with existing backdoor detectors by adding a single penalty term. Experiments show good backdoor detection results with low overhead.
>
> **SR1:** Thank you for the supportive comments about our work.
> ___
> **Weaknesses:**
>
> **W1:** The approach assumes access to a small but representative clean set per class, and may be sensitivity to domain shift, label noise.
>
> **WR1:** Thank you for this comment. Regarding domain shift, we now do an experiment wherein the clean samples are altered, either through color jitter or by Gaussian blurring.  Table 1 below shows that our method is pretty robust to this type of domain shift. Moreover, regarding label noise, we now also include an experiment in Table 2 below where the clean set is contaminated by mislabeling (considering mislabeling fractions up to 30 %).  The results show that MMBD-CSO detection accuracy is pretty robust to these ''insults'' applied to the clean set.
>
> **Table 1: Detection accuracy on clean data with domain shift.**
> |Domain shift|w/o|Color jitter|Guassian blur|
> |-|-|-|-|
> |BadNet|96|94|96|
> |ML-BadNet|70|70|74|
>
> **Table 2: Detection accuracy on clean data of different noise level.**
> |Noise level|0%|10%|20%|30%|
> |-|-|-|-|-|
> |BadNet|96|96|90|90|
> |ML-BadNet|70|72|68|66|
>
> Please refer to Appendix A.8 of our revision for more details.
> ___
> **W2:** As the motivation is based on backdoor features lying outside the intrinsic subspace, adaptive attacks that force overlap can escape it.
>
> **WR2:** Indeed, one should expect that the most challenging attack for CSO approaches to detect will involve backdoor triggers that contain intrinsic features of the target class of an attack.  This is why we included the Adaptive-Blend-2 experiment, with results in Table 5 (showing that our true detection rate is pretty high even for this challenging adaptive attack) and detailed description in the original Appendix A.2.
>
> But to more convincingly respond, the revised paper now gives more comprehensive Adaptive-Blend-2 experiments, involving larger blending ratios and sample specific triggers to make the attack even more challenging for CSO detectors. More specifically, on CIFAR-10, we now consider 'dog' as the backdoor target class, with the backdoor pattern a 16x16 pixel dog's face. During training, for each poisoned image, we randomly crop the backdoor to a 8x8 patch and blend this patch into the source class image. By using a different random cropping for each poisoned image, we are capturing _all_ of the dog's intrinsic features across the set of poisoned training images. We are also limiting the backdoor pattern to 8x8. At test time, we evaluate the attack success rate by blending a randomly cropped 8x8 backdoor trigger pattern. This adaptive attack should be more challenging for CSO detectors than the original one considered in the paper as it is learning all the dog's face features as the backdoor. Moreover, while we only considered a blending ratio of 0.2 in the original paper, in the revised paper we now consider blending ratios ranging from 0.2 all the way up to 0.8. As seen in Table 3 below, the cosine similarity between the backdoored source class image features (masked to capture intrinsic features of the target class) and the target class clean image features becomes larger as the blend ratio increases.  This is as one would expect. However, MMBD-CSO still achieves good detection accuracy even up to a blend ratio 0.8. This proves that our CSO method does work well against backdoor features that overlap with intrinsic features.
>
> **Table 3: Results for Adaptive-Blend-2 with different blend ratios.**
>
> | Blend Ratio → | 0.2 | 0.3 | 0.4 | 0.5 | 0.6 | 0.7 | 0.8 |
> |-|-|-|-|-|-|-|-|
> | Trigger-Intrinsic Feature Overlap | 0.40 | 0.40 | 0.54 | 0.52 | 0.60 | 0.53 | 0.64 |
> | Target Class Intrinsic Feature Overlap | 0.67 | 0.63 | 0.65 | 0.65 | 0.68 | 0.64 | 0.67 |
> | ASR (%) | 88.33 | 90.41 | 97.22 | 98.79 | 99.86 | 100.00 | 100.00 |
> | ACC (%) | 91.20 | 91.35 | 91.34 | 91.51 | 91.23 | 91.44 | 91.50 |
> | DA (%) | 100 | 100 | 100 | 100 | 96 | 96 | 90 |
>
> Please refer to Section 3.3 and Appendix A.3 of our revision for more details.

---

> ### Author Response · Authors · 2025-11-23
> **Author Response to Reviewer tvn1 (Part 2 of 2)**
>
> **W3:** Experiments use only moderate-scale backbones, it would be desirable to see results on larger modern models and vision transformers.
>
> **WR3:** We now include experiments on larger modern model VGG-16 [1] and vision transformer ViT [2]. As seen in Table 4 below, MMBD-CSO consistently achieves substantially higher detection accuracy compared with MMBD. These results further demonstrate the generalizability and robustness of our proposed approach across diverse neural network architectures.
>
> **Table 4: Detection accuracy on larger model architectures.**
> |Model|Attack|MMBD|MMBD-CSO|
> |-|-|-|-|
> |VGG-16|BadNet|68|92|
> | |ML-BadNet|0|36|
> |ViT|BadNet|42|78|
> | |ML-BadNet|0|20|
>
> Please refer to Appendix A.6 of our revision for more details.
>
> **Reference.**
> 1. Karen Simonyan and Andrew Zisserman. ''Very Deep Convolutional Networks for Large-Scale Image Recognition.'' ICLR, 2015.
> 2. Alexey Dosovitskiy,et al. ''An image is worth 16x16 words: Transformers for image recognition at scale.'' ICLR, 2021.
> ___
> **Questions:**
>
> **Q1:** How do CSO variant backdoor detectors depend on the number of images in the clean dataset per class? It would be helpful to better understand the relation between the choice of clean dataset and detector's performance.
>
> **QR1:** First, please note that our method used a much smaller number of clean samples than peer methods while still outperforming these methods.  Second, we did an ablation study on the size of the clean set in the original manuscript in Table 14 of the Appendix – this experiment shows that our method is quite effective even when there are even as few as 5 clean images per class.
> ___
> **Q2:** If an adaptive attacker explicitly aligns trigger features with the target class's intrinsic subspace, how can CSO be adapted?
>
> **QR2:** Please see **WR2** to your **W2** for a more detailed discussion.
> ___
> **Q3:** It is shown that sensitivity to lambda is modest, but values differ by detector. Is there a way to efficiently tune this hyperparameter for different datasets and detectors?
>
> **QR3:** We thank the reviewer for this comment. In fact, although we did not mention it in the original manuscript, we did use an automated approach for setting the hyperparameter lambda for each dataset and detector.  In particular, we set lambda to achieve balance between(approximately the same values for) the primal loss objective $\mathcal{L}$ and the CSO penalty, $C$.  For all our reported results, in all tables, lambda was set in this fashion (to balance, as much as possible, the L and C cost terms), which we found works quite well across datasets and detectors.  We now mention this in the revised manuscript on page 7, line 325-328.

---

### Official Review · Reviewer_ERPo · 2025-10-31

**Soundness:** 2
**Presentation:** 2
**Contribution:** 2
**Rating:** 4
**Confidence:** 3

**Summary:**

This paper proposes Class Subspace Orthogonalization (CSO), a plug-in framework designed to enhance the sensitivity of post-training backdoor detectors. The key idea is to suppress class-intrinsic features by orthogonalizing detector optimization away from each class’s intrinsic subspace, allowing the detector to focus on backdoor-related signals. CSO can be applied to existing detectors such as Neural Cleanse and MMBD, leading to significant improvements in detection accuracy and robustness across multiple datasets and attack types. The authors also introduce a Mixed-Label (ML) Attack that combines clean- and dirty-label poisoning to create more stealthy, harder-to-detect backdoors. Experiments on CIFAR-10, GTSRB, and Tiny-ImageNet show that CSO consistently improves sensitivity and maintains robustness even under adaptive attacks and limited clean data.

**Strengths:**

The paper is well-motivated and clearly addresses the sensitivity limitation of existing post-training backdoor detectors. The proposed Class Subspace Orthogonalization (CSO) is simple, detector-agnostic, and effective without modifying model architectures. It consistently improves detection accuracy and robustness across multiple datasets, detectors, and attack types. The introduction of the Mixed-Label (ML) Attack adds practical value by revealing new detection challenges. Experiments are comprehensive and well-analyzed, demonstrating both the effectiveness and generality of the proposed method.

**Weaknesses:**

- **Dependence on clean data**: CSO requires a small clean dataset to estimate class-intrinsic subspaces. However, prior work such as *Multidomain Active Defense: Detecting Multidomain Backdoor Poisoned Samples via All-to-All Decoupling Training without Clean Datasets* has shown that relying on clean data can lead to a high false-positive rate when detecting samples from out-of-distribution (OOD) domains. This raises concerns about CSO’s robustness and reliability in more realistic settings where truly clean data are not guaranteed.
- **Effectiveness against clean-label attacks remains unverified**: Although CSO is evaluated under mixed-label and partially clean-label settings (e.g., LC attack), it does not include experiments on purely clean-label backdoor attacks such as SIG or CLBD. It remains unclear whether CSO can effectively detect or mitigate fully clean-label poisoning.
- **No analysis on multi-trigger attacks**: The paper does not consider more complex multi-trigger or multi-target attack scenarios, which have been shown in recent works (e.g. M-to-n backdoor paradigm: A multi-trigger and multi-target attack to deep learning models; Poster: Multi-target & multi-trigger backdoor attacks on graph neural networks ) to significantly challenge post-training detection methods. Evaluating CSO under such settings would provide a more comprehensive understanding of its robustness.

- **Adaptive attack results are underemphasized and poorly integrated**: The evaluation of adaptive attacks (Adaptive-Blend and Adaptive-Blend-2) appears only briefly in Section 4.4 and the appendix, without quantitative results in the main tables or figures. Since adaptive robustness is a key claim of the paper, these results should be clearly reported in the main text with detailed metrics, variance, and comparisons. The current presentation makes it difficult to assess how much robustness CSO actually provides under adaptive conditions.

**Questions:**

1. The paper assumes the availability of a small clean dataset to estimate class-intrinsic subspaces. How sensitive is CSO’s performance to the size, quality, or domain shift of this clean data?
2. The proposed method is evaluated under mixed-label and partially clean-label attacks (e.g., LC, ML attack). Has CSO been tested against purely clean-label backdoor attacks such as SIG or CLBD, and how would it perform in those cases?
3. The paper does not analyze multi-trigger or multi-target attacks, which have been shown in recent studies to significantly challenge post-training detection methods. How might CSO handle such complex attack patterns?
4. Since CSO depends on class subspace estimation, could the method fail when the victim model’s representation is not linearly separable or when feature overlap occurs across classes?

---

> ### Author Response · Authors · 2025-11-23
> **Author Response to Reviewer ERPo (Part 1 of 3)**
>
> Thanks for your time and reviewing effort.
> ___
> **Summary:** This paper proposes Class Subspace Orthogonalization (CSO), a plug-in framework designed to enhance the sensitivity of post-training backdoor detectors...Experiments on CIFAR-10, GTSRB, and Tiny-ImageNet show that CSO consistently improves sensitivity and maintains robustness even under adaptive attacks and limited clean data.
>
> **Response:** Thank you for the accurate summary of our paper.
>
> **Strengths:** The paper is well-motivated and clearly addresses the sensitivity limitation of existing post-training backdoor detectors. The proposed Class Subspace Orthogonalization (CSO) is simple, detector-agnostic, and effective without modifying model architectures. It consistently improves detection accuracy and robustness across multiple datasets, detectors, and attack types. The introduction of the Mixed-Label (ML) Attack adds practical value by revealing new detection challenges. Experiments are comprehensive and well-analyzed, demonstrating both the effectiveness and generality of the proposed method.
>
> **Response:** Thank you for these highly supportive comments about our work.
> ___
> **Weaknesses:**
>
> **W1:** Dependence on clean data: CSO requires a small clean dataset to estimate class-intrinsic subspaces...This raises concerns about CSO’s robustness and reliability in more realistic settings where truly clean data are not guaranteed.
>
> **WR1:** Most post-training detectors rely on some clean data.  As shown by our experiments (see Tables 1 and 2 in the original manuscript) our method achieves better detection results than peer methods while at the same time making use of _far fewer_ clean samples.  For MMBD-CSO, MLBD-CSO, and NC-CSO we used just 10 clean examples per class.  This is a tiny amount of clean data, and much smaller than the amount needed by the BTI-DBF methods (250 samples per class). Note also that in Table 14 in the original manuscript we did an ablation study which shows that our method works quite well even when there are as few as 5 clean samples per class.  Finally, please note that in the revised paper Appendix A.8, we now also do experiments where the clean set has been corrupted, either by domain shift or by mislabeling -- the CSO results are pretty robust in the face of these ''distortions'' of the clean set. Please refer to **QR1** to your **Q1** and Appendix A.8 of our revision for more details.
> ___
> **W2:** Effectiveness against clean-label attacks remains unverified: Although CSO is evaluated under mixed-label and partially clean-label settings (e.g., LC attack), it does not include experiments on purely clean-label backdoor attacks such as SIG or CLBD. It remains unclear whether CSO can effectively detect or mitigate fully clean-label poisoning.
>
>  **W2:** Thank you for this comment. In Table 1 below, we now include experimental results for the SIG attack [1], and also two other clean label attacks -- Refool [2] and Narcissus [3].  We follow the default settings used in the original papers. The results confirm that our method is effective at detecting clean-label data poisoning attacks.
>
> **Table 1. Detection accuracy for more clean label backdoor attacks.**
>
> |CIFAR-10|Refool|Narcissus|SIG|
> |-|-|-|-|
> |MMBD|76|16|100|
> |MMBD-CSO|94|56|100|
>
> Please refer to Appendix A.5.3 of our revision for more details.
>
> **Reference.**
> 1. Mauro Barni, et al. ''A new backdoor attack in cnns by training set corruption without label poisoning." ICIP, 2019.
> 2. Y. Liu, et al. ''Reflection Backdoor: A Natural Backdoor Attack on Deep Neural Networks.'' ECCV, 2020.
> 3. Yi Zeng, et al. ''Narcissus: A practical clean-label backdoor attack with limited information.'' CCS, 2023.

---

> ### Author Response · Authors · 2025-11-23
> **Author Response to Reviewer ERPo (Part 2 of 3)**
>
> **W3:** No analysis on multi-trigger attacks: The paper does not consider more complex multi-trigger or multi-target attack scenarios, which have been shown in recent works (e.g. M-to-n backdoor paradigm: A multi-trigger and multi-target attack to deep learning models; Poster: Multi-target \& multi-trigger backdoor attacks on graph neural networks ) to significantly challenge post-training detection methods. Evaluating CSO under such settings would provide a more comprehensive understanding of its robustness.
>
> **WR3:** Thank you for this comment. As there is no open source code for ''M-to-n backdoor paradigm: A multi-trigger and multi-target attack to deep learning models'', and ''Poster: Multi-target \& multi-trigger backdoor attacks on graph neural networks'' focus on graph network domain, we reproduced MBTA [1], a multi-trigger attack that incorporates 10 different triggers into a single attack. We considered all-to-one MTBA and followed the default settings used in the original paper and provided in the open-sourced codes. 10 backdoored models are trained with randomly chosen target class, and we applied MMBD and MMBD-CSO for each model 5 times. MMBD achieves 90% detection accuracy, while MMBD-CSO achieves 98%, which demonstrates our method's resistance to multi-trigger attacks.
>
> Please refer to Appendix A.5.4 of our revision for more details.
>
> **Reference.**
> 1. Yige Li, et al. ''Multi-trigger backdoor attacks: More triggers, more threats.‘’ CoRR, 2024.
> ___
> **W4:** Adaptive attack results are underemphasized and poorly integrated.
>
> **WR4:**  Indeed, one should expect that the most challenging attack for CSO approaches to detect will involve backdoor triggers that contain intrinsic features of the target class of an attack.  This is why we included the Adaptive-Blend-2 experiment, with results in Table 5 and detailed description in Appendix A.2 in the original manuscript.  This table showed that our true detection rate is pretty high even for this challenging adaptive attack.
>
> But we agree with the reviewer both that: 1) a more convincing adaptive attack experiment is needed and 2) that this should be integrated into the main paper's text.
>
> To respond to the reviewer’s comments, in the paper revision we now do more comprehensive Adaptive-Blend-2 experiments, involving larger blending ratios and sample specific triggers to make the attack even more challenging for CSO detectors. More specifically, on CIFAR-10, we now consider 'dog' as the backdoor target class, with the backdoor pattern a 16x16 pixel dog's face. During training, for each poisoned image, we randomly crop the backdoor to a 8x8 patch and blend this patch into the source class image. By using a different random cropping for each poisoned image, we are capturing _all_ of the dog's intrinsic features across the collection of poisoned training images.  At the same time, we are limiting the backdoor pattern to 8x8 (16x16 would be too large, given the small size of CIFAR-10 images). At test time, we evaluate the attack success rate by blending a randomly cropped 8x8 backdoor trigger pattern. This adaptive attack should be more challenging for CSO detectors than the original one considered in the paper as it is learning all the dog's face features (all of the target class's intrinsic features) as the backdoor. Moreover, while we only considered a blending ratio of 0.2 in the original paper, in the revised paper we now consider blending ratios ranging from 0.2 all the way up to 0.8. As seen in Table 2 below, the cosine similarity between the backdoored source class image features (masked to capture intrinsic features of the target class) and the target class clean image features becomes larger as the blend ratio increases.  This is as one would expect. However, MMBD-CSO still achieves good detection accuracy even up to a blend ratio 0.8. This proves that our CSO method does work well against backdoor features that are similar to intrinsic features.
>
> **Table 2: Results for Adaptive-Blend-2 with different blend ratios.**
>
> | Blend Ratio → | 0.2 | 0.3 | 0.4 | 0.5 | 0.6 | 0.7 | 0.8 |
> |-|-|-|-|-|-|-|-|
> | Trigger-Intrinsic Feature Overlap | 0.40 | 0.40 | 0.54 | 0.52 | 0.60 | 0.53 | 0.64 |
> | Target Class Intrinsic Feature Overlap | 0.67 | 0.63 | 0.65 | 0.65 | 0.68 | 0.64 | 0.67 |
> | ASR (%) | 88.33 | 90.41 | 97.22 | 98.79 | 99.86 | 100.00 | 100.00 |
> | ACC (%) | 91.20 | 91.35 | 91.34 | 91.51 | 91.23 | 91.44 | 91.50 |
> | DA (%) | 100 | 100 | 100 | 100 | 96 | 96 | 90 |
>
>
> Please refer to Section 3.3 and Appendix A.3 of our revision for more details. We thank the reviewer again for prompting this more extensive adaptive attack experiment.

---

> ### Author Response · Authors · 2025-11-23
> **Author Response to Reviewer ERPo (Part 3 of 3)**
>
> **Questions:**
>
> **Q1:** The paper assumes the availability of a small clean dataset to estimate class-intrinsic subspaces. How sensitive is CSO’s performance to the size, quality, or domain shift of this clean data?
>
> **QR1:** Regarding the size of the small clean dataset, we again note that: i) our method used a much smaller number of clean samples than peer methods while still outperforming these methods and ii) We did an ablation study on the size of the clean set in the original manuscript in Table 14 of the Appendix – this experiment shows that our method is quite effective even when there are just 5 clean images per class. Regarding quality of the clean data, we now do an experiment in Table 3 where the clean set is contaminated by mislabeling (considering mislabeling fractions up to 30 %). Moreover, regarding domain shift, we now also include an experiment wherein the clean samples are altered, either through color jitter or by Gaussian blurring.  Table 4 shows that our method is pretty robust to this type of domain shift. The results show that MMBD-CSO detection accuracy is pretty robust to these ''insults'' applied to the clean set.
>
> **Table 3: Detection accuracy on clean data of different noise level.**
> |Noise level|0%|10%|20%|30%|
> |-|-|-|-|-|
> |BadNet|96|96|90|90|
> |ML-BadNet|70|72|68|66|
>
> **Table 4: Detection accuracy on clean data with domain shift.**
> |Domain shift|w/o|Color jitter|Guassian blur|
> |-|-|-|-|
> |BadNet|96|94|96|
> |ML-BadNet|70|70|74|
>
> Please refer to Appendix A.8 of our revision for more details.
> ___
> **Q2:** The proposed method is evaluated under mixed-label and partially clean-label attacks (e.g., LC, ML attack). Has CSO been tested against purely clean-label backdoor attacks such as SIG or CLBD, and how would it perform in those cases?
>
> **QR2:** Please refer to **WR2** to your **W2**, and Appendix A.5.3 in our revision for the evaluation against three additional clean-label attacks: SIG, Refool and Narcissus. The results confirm that our method is effective at detecting these additional clean-label data poisoning attacks.
> ___
> **Q3:** The paper does not analyze multi-trigger or multi-target attacks, which have been shown in recent studies to significantly challenge post-training detection methods. How might CSO handle such complex attack patterns?
>
> **QR3:** Please refer to **WR3** to your **W3** and Appendix A.5.4 in our revision for the evaluation against a multi-trigger attack MTBA, which demonstrates our method's resistance to multi-trigger attacks.
> ___
> **Q4:** Since CSO depends on class subspace estimation, could the method fail when the victim model’s representation is not linearly separable or when feature overlap occurs across classes?
>
> **QR4:**  This is a good question. The feature overlap across classes can be experimentally understood by measuring, for different attacks, the average rectified, masked feature cosine similarity, between samples from non-target classes and samples from the target class. We show these results for CIFAR-10 in Table 5 below. We also evaluate the average rectified, masked feature cosine similarity between _pairwise_ target class samples as a baseline for comparison.  The results show that the correlations between non-target class sample features and target class sample features are not small. In fact, these correlations, though smaller, are not much smaller than the correlations between sample features from the same (target) class.  This is in fact not so surprising, since different CIFAR-10 classes share many high-level attributes.  For example, dogs, cats, horses, and frogs all have eyes, and they all have legs.  Despite these ''feature overlaps'', MMBD-CSO (in Tables 1 and 2, as well as in many other tables in the Appendix in the original manuscript), is achieving strong detection results, and much better than existing baselines, against an array of attacks on this (CIFAR-10) data set.
>
> **Table 5: Source class intrinsic features and target class intrinsic feature overlap, and target class intrinsic feature overlap for different backdoor attacks.**
>
> |Attack→|BadNet|chess|1-pixel|blend|WaNet|IAD|LC|Bpp|
> |-|-|-|-|-|-|-|-|-|
> |Source Class, Target Class Intrinsic Feature Overlap|0.39|0.31|0.29|0.28|0.34|0.24|0.29|0.26|
> |Target Class Intrinsic Feature Overlap|0.56|0.50|0.71|0.52|0.61|0.74|0.77|0.73|
>
> Please refer to Appendix A.2.2 of our revision for more details. We thank the reviewer again for requesting this analysis, which further supports the CSO approach and improves our paper.

---

### Official Review · Reviewer_VdYG · 2025-10-31

**Soundness:** 3
**Presentation:** 1
**Contribution:** 2
**Rating:** 2
**Confidence:** 4

**Summary:**

The paper introduces Class Subspace Orthogonalization (CSO), a wrapper framework that attempts to improve existing backdoor detectors by forcing their search to be orthogonal to a class's intrinsic feature subspace, as learned from a tiny clean dataset. The central claim is that this isolates the backdoor signal, improving detection sensitivity and reducing false positives. The authors also propose a new mixed label (ML) attack that combines dirty-label and clean-label poisoning, primarily to demonstrate that CSO is necessary to detect such subtle, stealthy threats.

**Strengths:**

- The paper provides an interesting approach (CSO) to improve backdoor detection by penalizing detection features that correlate with intrinsic features of the original data.
- The paper proposes a new threat (mixed label attack) that combines dirty-label and clean-label backdoor to reduce collateral damage.
- Extensive results show that CSO improves the detection accuracy of base detection methods. They also demonstrate the stealthiness of mixed label attack under backdoor detection.

**Weaknesses:**

- The writing is incoherent and hard to follow. There are two separate contributions: CSO and mixed label attack, but I don't clearly see their connection. The motivation for mixed label attack is also unclear, the paper does not explain why adding clean-label backdoor helps reduce collateral damage. The purpose of mixed label attack is not mentioned in the conclusion.
- As mentioned in the adaptive attack section, this method does not work well against backdoor features that are similar to intrinsic features. Stealthy or clean-label backdoor possesses this property. Experimental results show that CSO yields low detection accuracy on stealthy attacks like Wanet or Bpp, clean-label attacks like label consistency, and one-to-one setting.
- The paper should evaluate other stealthy or clean label backdoor, such as Refool, Narcissus, Hidden Trigger Backdoor Attack, etc.

**Questions:**

- What is the main contribution of this paper, CSO or mixed label attack? What is their connection?

---

> ### Author Response · Authors · 2025-11-23
> **Author Response to Reviewer VdYG (Part 1 of 4)**
>
> Thanks for your time and careful reviewing effort.
>
> **Summary:** The paper introduces Class Subspace Orthogonalization (CSO), a wrapper framework that attempts to improve existing backdoor detectors by forcing their search to be orthogonal to a class's intrinsic feature subspace, as learned from a tiny clean dataset. The central claim is that this isolates the backdoor signal, improving detection sensitivity and reducing false positives. The authors also propose a new mixed label (ML) attack that combines dirty-label and clean-label poisoning, primarily to demonstrate that CSO is necessary to detect such subtle, stealthy threats.
>
> **Response:** Thank you for your accurate summary of the contribution of our work.
> ___
> **Strengths:**
>
> **S1:** The paper provides an interesting approach (CSO) to improve backdoor detection by penalizing detection features that correlate with intrinsic features of the original data.
>
> **SR1:** Thank you for acknowledging the originality of our work and finding it interesting.
> ___
> **S2:** The paper proposes a new threat (mixed label attack) that combines dirty-label and clean-label backdoor to reduce collateral damage.
>
> **SR2:** Thank you for acknowledging that the mixed label attack we proposed is also novel.
> ___
> **S3:** Extensive results show that CSO improves the detection accuracy of base detection methods. They also demonstrate the stealthiness of mixed label attack under backdoor detection.
>
> **SR3:** Thank you for acknowledging our extensive experimentation, which vindicates the detection performance benefits of our CSO methods.
> ___
> **Weaknesses:**
>
> **W1:** The writing is incoherent and hard to follow. There are two separate contributions: CSO and mixed label attack, but I don't clearly see their connection.
>
> **WR1:** It is troubling to us that the reviewer is saying they do not see the connection between CSO and the mixed label attack.  In the original manuscript we provided explicit motivation for introducing the mixed label attack. On page 2, 4th paragraph, we wrote ''Also, to more stringently test CSO’s effectiveness for backdoor detection, we also introduce a novel, mixed dirty/clean label X-to-X poisoning attack…In addition to yielding a more ''surgical'' backdoor mapping, this novel attack is also substantially harder to detect than  traditional dirty label attacks, as will be seen.'' Also, even in the paper abstract we stated that ''we also evaluate against a novel, mixed…attack that is more surgical and harder to detect than traditional dirty-label attacks.'' Finally, in summarizing our paper, above, this reviewer has stated: ''The authors also propose a new mixed label (ML) attack that combines dirty-label and clean-label poisoning, primarily to demonstrate that CSO is necessary to detect such subtle, stealthy threats.'' How can the reviewer state that they don't see the connection when the reviewer has precisely stated this connection in their summary of our paper?
>
> To make this connection as clear as possible in the revised manuscript, we have made the following changes:
>
> 1) In the abstract, last line: "**Moreover, to make the detection problem even more challenging**, we also evaluate against a novel mixed clean/dirty label...".
>
> 2) On page 2, 4th paragraph,: "Also, to more stringently test **the effectiveness of CSO and baseline methods**...we also introduce a novel, mixed dirty/clean..."
>
> 3) We have added a transition sentence at the beginning of Section 2.4: "Next, we introduce a novel, stealthy attack that will be used in our experiments as a significant detection challenge, both for CSO detection methods and for the baseline methods with which we will compare."
>
> 4) In the Conclusions, we now write: "Evaluation on the mixed label attack shows that existing detectors are inadequate, and CSO is necessary, to help detect subtle, stealthy threats."

---

> > ### Author Response · Authors · 2025-11-23
> > **Author Response to Reviewer VdYG (Part 2 of 4)**
> >
> > **W2:** The motivation for mixed label attack is also unclear, the paper does not explain why adding clean-label backdoor helps reduce collateral damage.
> >
> > **WR2:** The reviewer's question is tantamount to asking why supervising labels help to learn a concept/class in general (this is the fundamental premise of supervised learning itself, which requires no explanation).  By providing supervising examples from classes that are not source classes of an attack, which contain the backdoor trigger, and which are correctly labeled, we are teaching the model to learn to correctly classify (to these non-source classes) even in the presence of the backdoor trigger.   We essentially explained this in Section 2.4, where we wrote: “However, in our approach, the attacker performs an additional step by also embedding the backdoor trigger into samples from $\mathcal{K}$ not in $\mathcal{S}$ or $\mathcal{T}$ while keeping their original labels intact.  This aims to ensure that the model will only misclassify to $\mathcal{T}$ when the backdoor trigger is applied to samples from $\mathcal{S}$, i.e. that there is little to no collateral damage.”
> >
> > To further clarify, in the revised paper, we have amended this statement to read: ''However, in our approach, the attacker performs an additional step by also embedding the backdoor trigger into samples from $\mathcal{K}$ not in $\mathcal{S}$ or $\mathcal{T}$ while keeping their original labels intact.  This teaches the model to learn to only misclassify to $\mathcal{T}$ when the backdoor trigger is applied to samples from $\mathcal{S}$, i.e. that there is little to no collateral damage.''
> > ___
> > **W3:** The purpose of mixed label attack is not mentioned in the conclusion.
> >
> > **WR3:** In the conclusion, we wrote: ''We evaluated CSO-enhanced detectors...as well as against a novel mixed dirty/clean label attack, proposed here, that is significantly harder to detect than traditional dirty-label attacks.'' The implication of this sentence is exactly what the reviewer has stated in his summary of our paper: we introduced the mixed label attack primarily to demonstrate that CSO is necessary to detect subtle, stealthy threats.  In the revised paper, we have modified the conclusion statement as follows: ''We also evaluated against a novel mixed dirty/clean label attack, proposed here, that is harder to detect than traditional dirty-label attacks.  This evaluation shows that existing approaches are inadequate, and CSO is necessary, to help detect subtle, stealthy threats.''

---

> ### Author Response · Authors · 2025-11-23
> **Author Response to Reviewer VdYG (Part 3 of 4)**
>
> **W4:** As mentioned in the adaptive attack section, this method does not work well against backdoor features that are similar to intrinsic features.
>
> **WR4:** Indeed, one should expect that the most challenging attack for CSO approaches to detect will involve backdoor triggers that contain intrinsic features of the target class of an attack.  This is why we included the Adaptive-Blend-2 experiment, with results in Table 5 and detailed description in Appendix A.2 in the original manuscript.  This table in fact showed that our true detection rate is pretty high even for this challenging adaptive attack.
>
> But to more convincingly respond to the reviewer’s comment, in the paper revision we now do more comprehensive Adaptive-Blend-2 experiments, involving larger blending ratios and sample specific triggers to make the attack even more challenging for CSO detectors. More specifically, on CIFAR-10, we now consider 'dog' as the backdoor target class, with the backdoor pattern a 16x16 pixel dog's face. During training, for each poisoned image, we randomly crop the backdoor to a 8x8 patch and blend this patch into the source class image. By using a different random cropping for each poisoned image, we are capturing _all_ of the dog's intrinsic features across the collection of poisoned training images.  At the same time, we are limiting the backdoor pattern to 8x8 (16x16 would be too large, given the small size of CIFAR-10 images). At test time, we evaluate the attack success rate by blending a randomly cropped 8x8 backdoor trigger pattern. This adaptive attack should be more challenging for CSO detectors than the original one considered in the paper as it is learning all the dog's face features (all of the target class's intrinsic features) as the backdoor. Moreover, while we only considered a blending ratio of 0.2 in the original paper, in the revised paper we now consider blending ratios ranging from 0.2 all the way up to 0.8. As seen in Table 1 below, the cosine similarity between the backdoored source class image features (masked to capture intrinsic features of the target class) and the target class clean image features becomes larger as the blend ratio increases.  This is as one would expect. However, MMBD-CSO still achieves good detection accuracy even up to a blend ratio 0.8. This proves that our CSO method does work well against backdoor features that are similar to intrinsic features.
>
> **Table 1: Results for Adaptive-Blend-2 with different blend ratios.**
>
> | Blend Ratio → | 0.2 | 0.3 | 0.4 | 0.5 | 0.6 | 0.7 | 0.8 |
> |-|-|-|-|-|-|-|-|
> | Trigger-Intrinsic Feature Overlap | 0.40 | 0.40 | 0.54 | 0.52 | 0.60 | 0.53 | 0.64 |
> | Target Class Intrinsic Feature Overlap | 0.67 | 0.63 | 0.65 | 0.65 | 0.68 | 0.64 | 0.67 |
> | ASR (%) | 88.33 | 90.41 | 97.22 | 98.79 | 99.86 | 100.00 | 100.00 |
> | ACC (%) | 91.20 | 91.35 | 91.34 | 91.51 | 91.23 | 91.44 | 91.50 |
> | DA (%) | 100 | 100 | 100 | 100 | 96 | 96 | 90 |
>
> **W5:** Stealthy or clean-label backdoor possesses this property. Experimental results show that CSO yields low detection accuracy on stealthy attacks like Wanet or Bpp, clean-label attacks like label consistency, and one-to-one setting.
>
> **WR5:** Consider the experimental results in Table 1 in the original manuscript, for example, it is true that CSO methods have low absolute true detection rates on IAD, LC and Bpp attacks.  However, please note that each CSO variant is improving on its associated baseline detector.  For example, MMBD-CSO has detection rates of 84, 76, and 36 on these respective attacks, compared with detection rates of 76, 30, and 0 for the baseline MMBD method. In summary, while the absolute performance of CSO variants against these 3 attacks might not be perfect, the relative performance is quite strong, compared both against the baseline detectors and against other peer detectors (UNICORN, BTI-DBF, and BTI-DBF2).
>
> Finally, the detection rates in our result tables for all methods are all ''unusually'' low because we chose a lower poisoning rate (to make the detection problem as challenging as possible). When higher poisoning rates are chosen, all methods generally achieve higher true detection rates (Much higher poisoning rates were used in the original papers for the baseline methods, and much higher detection rates were reported in these original papers). We now demonstrate this for MMBD and MMBD-CSO in Table 2 below. Here, the poisoning rate was increased to 10% and the detection rates are _much_ higher on IAD, LC and Bpp than the results reported in our original Table 1. At the same time, MMBD-CSO is still achieving a detection advantage over MMBD -- at this higher poisoning rate, it is achieving excellent performance in detecting these 3 stealthy attacks, see Appendix A.7 in our revision.
>
> **Table 2. Detection accuracy for all-to-one WaNet, IAD, LC and Bpp attacks with higher poisoning rates of 10%.**
>
> |CIFAR-10|WaNet|IAD|LC|Bpp|
> |-|-|-|-|-|
> |MMBD|98|88|78|66|
> |MMBD-CSO|96|96|100|88|

---

> ### Author Response · Authors · 2025-11-23
> **Author Response to Reviewer VdYG (Part 4 of 4)**
>
> **W6:** The paper should evaluate other stealthy or clean label backdoor, such as Refool, Narcissus, Hidden Trigger Backdoor Attack, etc.
>
> **WR6:** In Table 3 below, we now evaluate against Refool [1], Narcissus [2], and SIG [3] attacks.  We followed the default settings used in the original papers. The results of these experiments show that MMBD-CSO makes improvement in detection accuracy over MMBD across these three additional clean label attacks.
>
> **Table 3. Detection accuracy for more clean label backdoor attacks.**
>
> |CIFAR-10|Refool|Narcissus|SIG|
> |-|-|-|-|
> |MMBD|76|16|100|
> |MMBD-CSO|94|56|100|
>
> Please refer to Appendix A.5.3 of our revision for more details.
>
> **Reference.**
> 1. Y. Liu, et al. ''Reflection Backdoor: A Natural Backdoor Attack on Deep Neural Networks.'' ECCV, 2020.
> 2. Yi Zeng, et al. ''Narcissus: A practical clean-label backdoor attack with limited information.'' CCS, 2023.
> 3. Mauro Barni, et al. ''A new backdoor attack in cnns by training set corruption without label poisoning." ICIP, 2019.
> ___
> Questions:
>
> **Q1:** What is the main contribution of this paper, CSO or mixed label attack? What is their connection?
>
> **QR1:** The reviewer wrote in their summary: ''The authors also propose a new mixed label (ML) attack that combines dirty-label and clean-label poisoning, primarily to demonstrate that CSO is necessary to detect such subtle, stealthy threats.'' Clearly, most of the developments in our paper are devoted to CSO.  We only concisely introduced the mixed label attack and then used it as a subtle attack to challenge existing and new (CSO) detectors.  From our original manuscript, the main contribution is quite clearly the CSO plug-and-play approach.  The mixed label attack is a secondary contribution. It is nevertheless a contribution. In response to the reviewer’s prior comment above, we have made the connection between the two contributions as explicit as possible in the revised manuscript.  To make as explicit as possible the fact that the primary contribution is the CSO approach, in the revised manuscript, last paragraph of page 2, we now state: ''(1) Our primary contribution is CSO, a novel framework...We also have the following secondary contributions: (2) We proposed a mixed clean/dirty label attack...The primary purpose is to challenge existing and new (CSO) detectors against a subtle, stealthy attack...''

---

### Official Review · Reviewer_FRMn · 2025-10-31

**Soundness:** 2
**Presentation:** 2
**Contribution:** 2
**Rating:** 2
**Confidence:** 4

**Summary:**

In the post-training backdoor detection, for instance the Neural Cleanse,  the defenders try to reconstruct a universal perturbation as a potential trigger signal, and then analyze their statistic information to determine which is the suspect class. However, how to determine the target class from these reconstructed perturbations is challenging. This paper tries to suppress these normal intrinsic features to improve the detection performance.

**Strengths:**

Strength:

1.	This paper considered the reverse-engineering based backdoor detection, and provide the systemic experiments to compare their performance with other works.

2.	The author proposed a regularization which can be easily used as a plugin to improve the performance of the existing reverse-engineering based methods, like NC, NNBD etc.

**Weaknesses:**

Weakness:

1.	In the second paragraph of page 2, the authors claimed that low poisoning ratio affects the reverse-engineering-based detector? From my humble perspective, whether the model is backdoored or not is the main factor.

2.	Moreover, Wang et al.2019 (Neural Cleanse) tries to f find the perturbation/trigger from the input space, not the feature space. But in Section 2.2.2, the authors cited it and claimed a soft mask identifying the intrinsic feature subspace. It is confusing.

3.	I fully understand the authors’ idea, i.e., first using Equ 4 to identify the intrinsic feature per class, and then exploiting that feature to penalize the reconstructed sample whose feature is alignment with it. Since the trigger signal will activate the different feature with the benign image, this idea can work well for common backdoor attack. However, once this attacker tries to reduce the feature difference between trigger and benign sample [1], this idea will fail. Therefore, I think this new defense idea can be easily bypassed when the attacker uses the regularization shown in [1] regardless the attack types.

[1] Bypassing backdoor detection algorithms in deep learning

**Questions:**

See my concerns in weaknesses

---

> ### Author Response · Authors · 2025-11-23
> **Author Response to Reviewer FRMn (Part 1 of 2)**
>
> Thanks for your time and careful reviewing effort.
>
> **Summary:** In the post-training backdoor detection, for instance NC, the defenders try to reconstruct a universal perturbation as a potential trigger signal, and then analyze their statistic information to determine which is the suspect class. However, how to determine the target class from these reconstructed perturbations is challenging. This paper tries to suppress these normal intrinsic features to improve the detection performance.
>
> **Response:** Yes, our approach suppresses intrinsic features in maximizing a given detection statistic.  We demonstrate the wide ''plug and play'' generality of our method (that it can be applied to modify many existing detectors), and that it improves over many existing detectors, both in detection sensitivity (true positives) and specificity (false positives).
> ___
> **Strengths:**
>
> **S1:** This paper considered the reverse-engineering based backdoor detection, and provide the systemic experiments to compare their performance with other works.
>
> **SR1:** We not only considered reverse-engineering based detection:  MMBD (and MLBD) are not reverse-engineering detectors – the input that maximizes margin or the logit will not necessarily (and not in practice) be a good estimate of the backdoor trigger.  It is simply the input that maximizes the detection statistic (margin or logit).
> ___
> **S2:** The author proposed a regularization which can be easily used as a plugin to improve the performance of the existing reverse-engineering based methods, like NC, MMBD etc.
>
> **SR2:** Indeed, our approach is ''plug and play'' and can be used to extend and to improve upon the performance of many existing detectors. We now explicitly refer to our method as a ''plug-and-play'' approach on page 1, 2, 5 and 10.
> ___
> **Weaknesses:**
>
> **W1:** In the second paragraph of page 2, the authors claimed that low poisoning ratio affects the reverse-engineering-based detector? From my humble perspective, whether the model is backdoored or not is the main factor.
>
> **WR1:**
> 1) In our experiments, we purposefully (to make the detection problem most challenging and thus to highlight the performance differences between different detectors) considered much lower poisoning rates than in prior publications. E.g., we considered BadNet at 0.1% poisoning rate while UNICORN & BTI-DBF considered BadNet at 5%.
>
> 2) The detection accuracies in original manuscript's Table 1 (also in our other tables) are much lower than the accuracies reported in the original papers for the baseline methods (where higher poisoning rates were used). For example, UNICORN reported 95% detection accuracy for BadNet with 5% poisoning rate, but with 0.1% poisoning rate the detection accuracy is only 54%; BTI-DBF reported 100% detection accuracy with 5% poisoning rate, but with 0.1% poisoning rate the detection accuracy is only 72%. So we show that the poisoning rate does indeed have a significant influence on the effectiveness of post-training detectors. Though we considered low poisoning rates, we did **not** significantly compromise attack effectiveness: the attack success rates (Tables 8-13) are mostly above 90%. We make these points more explicitly in the revised manuscript in page 7, line 358-364.
>
> 3) In Table 1 below we now show that when the poisoning rate is increased for some of the most challenging attacks (WaNet, IAD, LC, and Bpp), MMBD-CSO achieves excellent (absolute) detection accuracy, and much better than its performance for the unusually low poisoning rates (reported in Table 1 in the original manuscript). These results do suggest that the poisoning rate has a strong influence on a given detector's performance. See Appendix A.7 of our revision for more details.
>
> **Table 1. Detection accuracy for all-to-one WaNet, IAD, LC and Bpp attacks at 10% poisoning rate.**
> |CIFAR-10|WaNet|IAD|LC|Bpp|
> |-|-|-|-|-|
> |MMBD|98|88|78|66|
> |MMBD-CSO|96|96|100|88|

---

> > ### Author Response · Authors · 2025-11-23
> > **Author Response to Reviewer FRMn (Part 2 of 2)**
> >
> > **W2:** Moreover, Wang et al.2019 (NC) tries to find the perturbation/trigger from the input space, not the feature space. But in Section 2.2.2, the authors cited it and claimed a soft mask identifying the intrinsic feature subspace. It is confusing.
> >
> > **WR2:** In our original manuscript, we did make it explicit that ''NC-CSO soft-masks both the input and the embedded ($S\_a$) feature space.'' But we agree that we could do a better job of explaining that these two different maskings are being used for different, complementary purposes. The soft-masking of the input (image) is being performed to estimate the spatial support of the backdoor trigger pattern (as in the original NC) – to localize a possible backdoor trigger pattern. By contrast, the soft-masking in embedded feature space applied by NC-CSO is used to estimate a putative target class’s intrinsic features (defined on this spatially localized support) – we orthogonalize with respect to these intrinsic, embedded features in maximizing the NC detector’s objective function. If we did not perform NC’s input feature masking, NC-CSO would not achieve spatial localization of estimated backdoor trigger patterns. And feature masking in embedded space is needed to identify a class's intrinsic features (with respect to which we orthogonalize). These two feature maskings are complementary, and both are necessary, within NC-CSO. We now make these points more clearly on page 5, line 249-252.
> > ___
> > **W3:** ...once this attacker tries to reduce the feature difference between trigger and benign sample [1], this idea will fail. Therefore, I think this new defense idea can be easily bypassed when the attacker uses the regularization shown in [1] regardless the attack types.
> >
> > **WR3:** Indeed, one should expect that the most challenging attack for CSO approaches to detect will involve backdoor triggers that contain intrinsic features of the target class of an attack. This is why we included the Adaptive-Blend-2 experiment, with results in Table 5 in the original manuscript. This table shows that our true detection rate is pretty high even for this challenging adaptive attack.
> >
> > But to respond further to the reviewer’s comment, in the paper revision we now do more comprehensive Adaptive-Blend-2 experiments, involving larger blending ratios and sample specific triggers to make the attack even more challenging for CSO detectors. More specifically, on CIFAR-10, we now consider 'dog' as the backdoor target class, with the backdoor pattern a 16x16 pixel dog's face. During training, for each poisoned image, we randomly crop the backdoor to a 8x8 patch and blend this patch into the source class image. By using a different random cropping for each poisoned image, we are capturing _all_ of the dog's intrinsic features across the collection of poisoned training images. At the same time, we are limiting the backdoor pattern to 8x8. At test time, we evaluate the attack success rate by blending a randomly cropped 8x8 backdoor trigger pattern. This adaptive attack should be more challenging for CSO detectors than the original one considered in the paper as it is learning all the dog's face features (all of the target class's intrinsic features) as the backdoor. Moreover, while we only considered a blending ratio of 0.2 in the original paper, in the revised paper we now consider blending ratios ranging from 0.2 all the way up to 0.8. As seen in Table 2 below, the cosine similarity between the backdoored source class image features (masked to capture intrinsic features of the target class) and the target class clean image features becomes larger as the blend ratio increases. This is as one would expect. However, MMBD-CSO still achieves good detection accuracy even up to a blend ratio 0.8.
> >
> > **Table 1: Results for Adaptive-Blend-2 with different blend ratios.**
> >
> > | Blend Ratio → | 0.2 | 0.3 | 0.4 | 0.5 | 0.6 | 0.7 | 0.8 |
> > |-|-|-|-|-|-|-|-|
> > | Trigger-Intrinsic Feature Overlap | 0.40 | 0.40 | 0.54 | 0.52 | 0.60 | 0.53 | 0.64 |
> > | Target Class Intrinsic Feature Overlap | 0.67 | 0.63 | 0.65 | 0.65 | 0.68 | 0.64 | 0.67 |
> > | ASR (%) | 88.33 | 90.41 | 97.22 | 98.79 | 99.86 | 100.00 | 100.00 |
> > | ACC (%) | 91.20 | 91.35 | 91.34 | 91.51 | 91.23 | 91.44 | 91.50 |
> > | DA (%) | 100 | 100 | 100 | 100 | 96 | 96 | 90 |
> >
> > In addition, we implemented Bypassing attack [1] on ten BadNet baseline models with randomly chosen target labels and then jointly trained the backdoored model with the discriminator network to suppress feature differences between backdoored and clean samples. MMBD detected no backdoor (DA = 0%), while MMBD-CSO achieved detection accuracy of 40%, demonstrating a huge improvement over the baseline MMBD method.
> >
> > See Section 3.3, Appendix A.3 and A.5.2 of our revision for more details.
> >
> > **Refererence.**
> > 1. Reza Shokri et al. ''Bypassing backdoor detection algorithms in deep learning.'' EuroS&P, 2020.

---

> > > ### Comment · Reviewer_FRMn · 2025-11-25
> > >
> > > Thanks for the authors' reply. For the Adaptive blend, the feature of the poisoned and benign are overlapping only when the poisoning ratio is tiny. Once the poisoning ratio is large or training super-long time (leading to a high ASR), the feature is still well separative. Therefore, the good performance against Adaptive blend cannot eliminate my concerns. Moreover, the proposed work cannot fully solve the ref[1], which further prove my concerns.

---

> > > > ### Author Response · Authors · 2025-11-27
> > > > **Author Response to Reviewer FRMn's new comment (Part 1 of 1)**
> > > >
> > > > Thanks for your comment. Please note that, in our revision's Adaptive-Blend-2 results table, the columns stand for different blend ratios, **not** poisoning rates(ratios). Blend ratio determines how the backdoor pattern is incorporated into a clean source-class example, while poisoning rate is the ratio of the amount of poisoned examples to total number of examples in the training dataset. That is, for blend ratio $\alpha = 0.2$, the backdoor pattern $p$ with a location mask $m$ is blended into a source image $x$ by a ratio of 0.2, so we have the poisoned source image as $(1-m)\odot x + \alpha *m\odot p + (1-\alpha)*m\odot x$. The poisoning rate we used for all experiments in Adaptive-Blend-2 is _fixed_ at 5%, which is not large compared to published results. Also, it is the **lowest** poisoning rate for Adaptive-Blend-2 to succeed at blend ratio of 0.2. Specifically, if we lower the poisoning rate from 5% to 4%, the attack success rate will be lowered from 88.33% to 65.2%. On the other hand, as what we've stated in the original manuscript, we trained all models for 100 epochs, which is not ''training super-long time''. We are using the target class discriminative features as our backdoor pattern, and pushing the limit of poisoning rate as low as possible to make the attack succeed, without ''large poisoning ratio'' or ''super-long training time''.
> > > > Furthermore, as seen in Adaptive-Blend-2 results table, the feature overlap of the poisoned and benign images is increasing as the blend ratio $\alpha$ increases, which is what one would expect, because for each poisoned image, we are blending more backdoor pattern $p$ into the source image $x$. This implies that by increasing $\alpha$, we are making the attack more stealthy to (difficult to detect by) our CSO method by breaking the latent separability assumption.
> > > >
> > > > For detection performance against [1], we want to emphasize that our detector MMBD-CSO demonstrates a substantial improvement over the baseline MMBD method. CSO is indeed guiding the baseline method toward a better optimization direction. This proves that CSO remains effective even when an attack specifically breaks the latent separability assumption.
> > > >
> > > > However, we understand that the reviewer still has concerns about the absolute performance of MMBD-CSO against [1]. To further demonstrate MMBD-CSO's detection ability against this attack, we now evaluate [1] on BTI-DBF [2] and UNICORN [3]. Both detectors are state-of-the-art defense methods. BTI-DBF has a detection accuracy of 22%, and UNICORN has a detection accuracy of 36%. So, even though MMBD-CSO's performance is not perfectly high, it does achieve competitive performance compared with these two representative approaches.
> > > >
> > > > **References:**
> > > > 1. Reza Shokri et al. ''Bypassing backdoor detection algorithms in deep learning.'' EuroS&P, 2020.
> > > > 2. Xiong Xu et al. ''Towards reliable and efficient backdoor trigger inversion via decoupling benign features." ICLR, 2024.
> > > > 3. Zhenting Wang et al. ''UNICORN: A Unified Backdoor Trigger
> > > > Inversion Framework." ICLR, 2023.

---

### Official Review · Reviewer_236s · 2025-11-01

**Soundness:** 3
**Presentation:** 3
**Contribution:** 2
**Rating:** 4
**Confidence:** 3

**Summary:**

To tackle the problem that the performance of backdoor attack detector deteriorates when some untargetted classes exhibit similarly extreme distributions, the authors propose a method that identifies and orthogonalizes intrinsic features, and applies the orthogonalization to existing backdoor defenses, achieving promising results.

**Strengths:**

(1) The method has a very reasonable motivation, and is grounded in theory.

(2) The experiment is convincing: it is not grounded in specifically curated datasets that requires very strict feature distribution difference but still outperforms baselines.

(3) The presentation of the paper is clear.

(4) The authors propose many variants of how the intrinsic features can be integrated into existing backdoor detectors.

**Weaknesses:**

Strength (4) somehow also becomes the weakness - I am mostly aware of the applicability of the method: if different integration methods have to be applied for different detector, how far can it go? What if there are more powerful detectors coming around and how hard is it to adapt your method?

Taking intrinsic features into account is nothing new in the community, and I suggest the authors to stress the major contribution of your work w.r.t. other existing methods.

**Questions:**

See weakness.

---

> ### Author Response · Authors · 2025-11-23
> **Author Response to Reviewer 236s (Part 1 of 1)**
>
> Thanks for your time and careful reviewing effort.
>
> **Summary:** To tackle the problem that the performance of backdoor attack detector deteriorates when some untargeted classes exhibit similarly extreme distributions, the authors propose a method that identifies and orthogonalizes intrinsic features, and applies the orthogonalization to existing backdoor defenses, achieving promising results.
>
> **Response:** Thank you for acknowledging that our method achieves promising results.  In fact, we demonstrated that it significantly boosts the performance of well-known detectors (both in terms of true detections and false positives) and outperforms all existing post-training detectors of which we are aware.
> ___
> **Strengths:**
>
> **S1:** The method has a very reasonable motivation, and is grounded in theory.
>
> **SR1:** Thank you for acknowledging the technical soundness of our approach.
> ___
> **S2:** The experiment is convincing: it is not grounded in specifically curated datasets that requires very strict feature distribution difference but still outperforms baselines.
>
> **SR2:** Thank you for this very supportive comment about the soundness of our experimentation.
> ___
> **S3:** The presentation of the paper is clear.
>
> **SR3:** Thank you for acknowledging the technical clarity of our paper.
> ___
> **S4:** The authors propose many variants of how the intrinsic features can be integrated into existing backdoor detectors.
>
> **SR4:**
> Yes, our CSO method can "plug and play" with many existing detectors to boost their performance if they optimize detection statistics that rely on intrinsic features.
> ___
> **Weaknesses:**
>
> **W1:** Strength (4) somehow also becomes the weakness - I am mostly aware of the applicability of the method: if different integration methods have to be applied for different detector, how far can it go? What if there are more powerful detectors coming around and how hard is it to adapt your method?
>
> **WR1:** Even in the 4 CSO variants we have introduced in the paper we are giving a good glimpse of ''how far'' our plug-and-play approach can go.  Note that NC optimizes over the estimated mask and trigger, PT-RED optimizes over the estimated (additive) trigger, and MMBD and MLBD optimize over the inputs to the model. CSO applies to all 4 of these detectors, which are optimizing over very different quantities, in generating detection statistics.  Note also that the detector objective function used in NC and in PT-RED, based on average cross entropy loss over a clean set, is very different from the margin and logit objective functions of MMBD, and MLBD – and yet we demonstrate that CSO can be naturally applied to all these detectors, with their quite different detector objective functions.  Although we did not explicate this in our paper, it is also straightforward to produce CSO variants of UNICORN and BTI-DBF – we expect that CSO would provide similar benefits to these detectors as it provides to NC, PT-RED, MMBD, and MLBD. Again, so long as an existing detector is optimizing a detection statistic that relies on intrinsic features, CSO should in principle be applicable and should improve upon this existing detector.  We now make these points more explicitly in the revised manuscript (page 5, line 265-269) to make it clearer ''how far [our method] can go''.
> ___
> **W2:** Taking intrinsic features into account is nothing new in the community, and I suggest the authors to stress the major contribution of your work w.r.t. other existing methods.
>
> **WR2:** In the above comment, the reviewer has not identified any existing detection frameworks that closely resemble our “plug and play” CSO approach.  This is the main novelty, which we emphasize throughout our paper (in the revised paper, we explicitly call out CSO as the primary contribution of the paper (page 2, line 106-107)).
>
> Also, it is indeed true that masking to focus on intrinsic features is used in several prior detectors (NC, FeatureRE, UNICORN, and BTI-DBF).  In Section 2.2.2 we acknowledged this, citing the FeatureRE, UNICORN, BTI-DBF works.  We also discussed these works in Section 4.1. In the revised paper, both in Section 2.2.2. and in Section 4.1 we acknowledge that our use of feature masking is indeed inspired by these prior works. At the same time, our class-dependent feature masking is different from the global feature masking used in these prior works.  In Appendix A.4 Table 15 of the original manuscript, we are showing that CSO with class-dependent masking achieves better results than a CSO variant that uses global masking. Moreover, our methods (with class-dependent masking) are substantially outperforming existing detectors that employ global feature masking (NC, UNICORN, and BTI-DBF).  We now make these points more explicitly in the Section 4.1 of our revision (page 10, line 508-511).

---

### Official Review · Reviewer_WUb8 · 2025-11-01

**Soundness:** 3
**Presentation:** 3
**Contribution:** 3
**Rating:** 6
**Confidence:** 3

**Summary:**

This paper introduces a framework called Class Subspace Orthogonalization (CSO) to enhance post-training backdoor detection sensitivity, and also proposes a new mixed-label backdoor attack to challenge detectors. The key idea is to address two failure modes of existing detectors: (1) cases where a benign class naturally appears as an outlier in detection statistics, and (2) cases where the backdoor trigger is subtle relative to normal class features. The authors observe that a backdoored target class contributes to a detector’s statistic via both its normal (intrinsic) features and the trigger, whereas non-target classes contribute only intrinsic features. Thus, if one can suppress the intrinsic class features during detection, any remaining strong signal would likely come from a trigger (if present). Based on this insight, the paper formulates a constrained optimization that maximizes a chosen backdoor detection statistic for each class while enforcing orthogonality to that class’s feature subspace. This CSO approach can be plugged into a wide variety of detectors to guide them toward backdoor cues and away from benign class patterns.

**Strengths:**

- Significant Boost in Sensitivity: The most evident strength of CSO is how it noticeably improves the sensitivity of backdoor detection. By removing the “noise” of normal class features, detectors become capable of catching very subtle backdoors that might have previously gone unnoticed. The paper’s introduction clearly articulates this benefit: for the backdoor target class, even if the trigger effect was weak, suppressing intrinsic features lets that weak signal stand out; for non-target classes, whose signals are purely intrinsic, the suppression drastically lowers their scores. This mechanism directly addresses the problem of missed detections. In the results, this translated to far higher detection accuracy, meaning fewer backdoored models would slip by. In practical terms, CSO could be the difference between an undetected Trojan model in deployment and one that gets caught before causing damage.
- Universality and Flexibility: CSO is designed as a plug-and-play framework. It’s not a detector by itself but rather a module that augments existing detectors. This is a great strength because it means CSO’s idea can be applied broadly without reinventing the wheel for each new detection method. The authors demonstrate integration with multiple algorithms (NC, PT-RED, MMBD, etc.) and mention it can be seamlessly integrated with other detectors too. Developers of backdoor defenses can adopt CSO’s penalty term in their own methods relatively easily. The fact that it worked across various types of detectors (anomaly-based, optimization-based, pairwise trigger search, etc.) shows its generality. Such flexibility ensures that CSO’s benefits are not limited to a niche case but can impact the whole landscape of post-training detection. Future detectors could include a “CSO step” as a standard to enhance their reliability.
- Reduced False Alarms: Increasing sensitivity often comes at the cost of more false positives, but CSO managed to improve true positive detection while keeping false positives low. This is a strong advantage – a detector that cries wolf too often will not be adopted in practice. CSO’s focus on differentiating trigger vs. intrinsic features is precisely what allows this balance: it filters out benign anomalies (like a naturally high-confidence class) which might fool a naive detector, thereby avoiding false alarms. The data showed CSO variants had equal or fewer false positives than comparable methods. For example, if a certain class in a clean model was particularly distinct, a normal detector might flag it wrongly; CSO ensures that unless there is a feature outside the normal subspace giving an abnormal boost, it won’t flag. This property makes CSO-enhanced detectors more trustworthy in real deployments.

**Weaknesses:**

### Assumption of Feature Separability
The core assumption of CSO is that the backdoor trigger introduces features that lie outside the normal feature subspace of the target class. While generally reasonable (triggers are usually patterns unrelated to the class, like a sticker on a stop sign), one can conceive of cases where this doesn’t hold. An attacker could choose a trigger that is a feature native to the target class. For example, suppose the target class is dogs, and the attacker’s trigger is “add pointy ears” – many dogs naturally have pointy ears, so this trigger might actually lie within the dog feature distribution. In such a case, enforcing orthogonality to the dog subspace would also filter out the trigger signal, potentially making the backdoor undetectable by CSO. In other words, if the trigger is not truly an orthogonal add-on but overlaps with intrinsic features, CSO could struggle. The authors’ adaptive attack Adaptive-Blend-2 is a step in this direction (blending target data with source-trigger data so the trigger is partially intrinsic). This is somewhat a worst-case scenario, but it’s a limitation: CSO works best when there is a clear distinction between what’s normal for a class and what’s introduced by the trigger.

**Questions:**

1. Quantifying “trigger–intrinsic” overlap (does CSO’s key assumption hold?) How separable are trigger features from intrinsic class features in practice? If overlapping exists, where does the overlap arise (layer-wise)?
2. It could be better to explore the effectiveness of the proposed defense on distribution-preserving backdoor attacks. e.g. [a], which intentionally induces distribution overlapping between backdoor samples and clean samples, to validate the generality of the intuition behind CSO.
[a] Distribution Preserving Backdoor Attack in Self-supervised Learning

---

> ### Author Response · Authors · 2025-11-23
> **Author Response to Reviewer WUb8 (Part 1 of 2)**
>
> Thank you for your time and careful reviewing effort.
> ___
> **Summary.** The reviewer provides an accurate summary of our core idea: a backdoored target class contributes to a detector’s statistic via both intrinsic class features and the trigger, whereas non-target classes contribute only intrinsic features. Thus, suppressing intrinsic features amplifies any remaining trigger signal, enabling more sensitive backdoor detection. As noted, our class subspace orthogonalization (CSO) module can be plugged into a wide range of existing detectors to guide them toward backdoor cues and away from benign class patterns.
>
> **Response.** We appreciate the reviewer’s close reading of our paper, as reflected by this accurate summary.
> ___
> **Strengths:**
>
> **S1:** Significant boost in sensitivity. The reviewer highlights that CSO noticeably improves detection sensitivity by removing the ''noise'' of intrinsic features, thereby allowing detectors to capture subtle triggers. This indeed translates into substantially higher detection accuracy.
>
> **Response:** Thank you for this highly supportive assessment.
> ___
> **S2:** Universality and flexibility. The reviewer emphasizes that CSO is a plug-and-play module that augments existing detectors rather than replacing them. This is precisely one of our main design goals.
>
> **Response:** We appreciate this insight; in the revised version, we explicitly refer to CSO as a plug-and-play approach (pages 1, 2, 5, 10).
> ___
> **S3:** Reduced false alarms. The reviewer notes that CSO improves sensitivity while keeping false positives low, increasing detector reliability.
>
> **Response:** We thank the reviewer for this positive observation—indeed, CSO reduces false alarms compared to peer baselines.
> ___
> **Weaknesses:**
>
> **W1:** Assumption of Feature Separability. The core assumption of CSO is that the backdoor trigger introduces features that lie outside the normal feature subspace of the target class. While generally reasonable (triggers are usually patterns unrelated to the class, like a sticker on a stop sign), one can conceive of cases where this doesn’t hold. An attacker could choose a trigger that is a feature native to the target class...
>
> **WR1:** Indeed, one should expect that the most challenging attack for CSO approaches to detect will involve backdoor triggers that contain intrinsic features of the target class of an attack.  This is why we included the Adaptive-Blend-2 experiment, with results in Table 5 and detailed description in Appendix A.2 in the original manuscript.  This table shows that our true detection rate is pretty high even for this challenging adaptive attack.
>
> To respond to the reviewer’s comment, to go beyond "Adaptive-Blend-2 is a step in this direction", in the paper revision we now do a more comprehensive Adaptive-Blend-2 experiment, involving larger blending ratios and sample specific triggers, to make the attack even more challenging for CSO detectors. More specifically, on CIFAR-10, we now consider 'dog' as the backdoor target class, with the backdoor pattern a 16x16 pixel dog's face. During training, for each poisoned image, we randomly crop the backdoor to a 8x8 patch and blend this patch into the source class image. By using a different random cropping for each poisoned image, we are capturing _all_ of the dog's intrinsic features across the collection of poisoned training images. At the same time, we are limiting the backdoor pattern to 8x8 (16x16 would be too large, given the small size of CIFAR-10 images). At test time, we evaluate the attack success rate by blending a randomly cropped 8x8 backdoor trigger pattern. This adaptive attack should be more challenging for CSO detectors than the original one considered in the paper as it is learning all the dog's face features (all of the target class's intrinsic features) as the backdoor. Moreover, while we only considered a blending ratio of 0.2 in the original paper, in the revised paper we now consider blending ratios ranging from 0.2 all the way up to 0.8. As seen in Table 1, the cosine similarity between the backdoored source class image features (masked to capture intrinsic features of the target class) and the target class clean image features becomes larger as the blend ratio increases. This is as one would expect. However, MMBD-CSO still achieves strong detection accuracy even up to a blend ratio of 0.8.
>
> **Table 1: Results for Adaptive-Blend-2 with different blend ratios.**
>
> | Blend Ratio → | 0.2 | 0.3 | 0.4 | 0.5 | 0.6 | 0.7 | 0.8 |
> |-|-|-|-|-|-|-|-|
> | Trigger-Intrinsic Feature Overlap | 0.40 | 0.40 | 0.54 | 0.52 | 0.60 | 0.53 | 0.64 |
> | Target Class Intrinsic Feature Overlap | 0.67 | 0.63 | 0.65 | 0.65 | 0.68 | 0.64 | 0.67 |
> | ASR (%) | 88.33 | 90.41 | 97.22 | 98.79 | 99.86 | 100.00 | 100.00 |
> | ACC (%) | 91.20 | 91.35 | 91.34 | 91.51 | 91.23 | 91.44 | 91.50 |
> | DA (%) | 100 | 100 | 100 | 100 | 96 | 96 | 90 |
>
> We have provided more details in Section 3.3 and Appendix A.3 of our revision.

---

> ### Author Response · Authors · 2025-11-23
> **Author Response to Reviewer WUb8 (Part 2 of 2)**
>
> Questions:
>
> **Q1:** Quantifying “trigger–intrinsic” overlap (does CSO’s key assumption hold?) How separable are trigger features from intrinsic class features in practice? If overlapping exists, where does the overlap arise (layer-wise)?
>
> **QR1:** This is a good question. ''Trigger–intrinsic'' overlap can be experimentally assessed by measuring the average rectified, masked feature cosine similarity, in various layers, between samples from non-target classes that contain the backdoor trigger and samples from the target class.  We now include this assessment in Table 2 below. The results show that the feature correlations between samples (without the trigger) from the same (target) class are larger than the feature correlations between source class samples with the trigger and target class samples (without the trigger).
> This is as one would expect, and is supportive of CSO's key assumption --
> that the intrinsic feature overlap between source class samples with the trigger and target class samples is low.
> In terms of the layer, we can see that, for most of the attacks (excepting IAD and Bpp), the trend is that the correlations in deeper layers (13 and 17) are smaller than in the earlier layers.  This gives some empirical support to our choice of a deep layer for measuring the CSO penalty. We thank the reviewer for requesting this analysis, which further supports the CSO approach and improves our paper.
>
> **Table 2. Layerwise trigger-intrinsic feature overlap and target class intrinsic feature overlap for different backdoors.**
> |Attack|Layer|Trigger–Intrinsic Feature Overlap|Target Class Intrinsic Feature Overlap|
> |-|-|-|-|
> |**BadNet**|5th|0.38|0.46|
> ||9th|0.26|0.32|
> ||13th|0.21|0.33|
> ||17th|0.32|0.56|
> |**chess**|5th|0.40|0.40|
> ||9th|0.30|0.32|
> ||13th|0.21|0.31|
> ||17th|0.29|0.50|
> |**1-pixel**|5th|0.40|0.42|
> ||9th|0.31|0.36|
> ||13th|0.28|0.39|
> ||17th|0.27|0.71|
> |**blend**|5th|0.39|0.45|
> ||9th|0.26|0.32|
> ||13th|0.22|0.34|
> ||17th|0.21|0.52|
> |**WaNet**|5th|0.40|0.44|
> ||9th|0.48|0.53|
> ||13th|0.53|0.56|
> ||17th|0.33|0.61|
> |**IAD**|5th|0.23|0.26|
> ||9th|0.49|0.52|
> ||13th|0.53|0.59|
> ||17th|0.58|0.74|
> |**LC**|5th|0.40|0.51|
> ||9th|0.26|0.33|
> ||13th|0.17|0.28|
> ||17th|0.22|0.77|
> |**Bpp**|5th|0.17|0.28|
> ||9th|0.53|0.55|
> ||13th|0.46|0.58|
> ||17th|0.63|0.73|
>
> Please refer to Appendix A.2.1 of our revision for more details.
> ___
> **Q2:** It could be better to explore the effectiveness of the proposed defense on distribution-preserving backdoor attacks. e.g. [a], which intentionally induces distribution overlapping between backdoor samples and clean samples, to validate the generality of the intuition behind CSO.
>
> [a] Distribution Preserving Backdoor Attack in Self-supervised Learning
>
> **QR2:** Thank you for this suggestion. In the revised manuscript, we now include an experimental evaluation of CSO against distribution-preserving backdoor attacks (DRUPE) in Appendix A.5.1.
> We reproduced the DRUPE attack with its official code under the default settings, where the encoder is pretrained on CIFAR-10 and downstream classifiers are trained on GTSRB. We generated five backdoored downstream models with different reference inputs per model.
> MMBD achieved a detection accuracy of only 4%, while MMBD-CSO achieved 60%, demonstrating a huge improvement over the baseline MMBD method for this challenging, advanced attack.
> ___
> Thanks again for your supportive comments about our work.

---

### Meta-Review · Area_Chair_g3GT · 2026-01-09

**Summary:**

The paper proposed Class Subspace Orthogonalization (CSO), a framework that can improve existing backdoor detectors, such as Neural Cleanse, by guiding their search orthogonally to a class's intrinsic feature subspace, using a tiny clean dataset. This strategy helps isolate the backdoor signal, improving detection while reducing false positives. The paper also proposes a mixed label attack, which combines dirty-label and clean-label poisoning, to demonstrate the utility of CSO against this stealthy threat.

This paper has received 6 reviews, which triggered a significant effort from the authors, something that I take into consideration very carefully. There are several concerns about the paper, but the main ones include (1) additional evaluation against various types of baselines, such as clean label, latent-separation mitigation, multi-target/trigger attacks, (2) experimental evaluation details and assumptions (e.g., clean image, poisoning ratios, etc…), (3) additional analysis on adaptive attacks. The responses from the author have addressed most of these issues but I think that the paper is still not yet sufficient.

While the paper extensively evaluated against several backdoor types, it should also provide insight into the limitations of the method, which is still limited in this work. The paper should include more rigorous empirical studies against backdoor risks that can potentially jeopardize the defense (the defense does not necessarily need to perform well in these scenarios); i.e., feature/latent separation focused attacks and multi-target/multi-trigger attacks. Qi et al. (2023) is not the only type of attack focusing on latent separation, and there are multiple target attacks such as  Xue et al. (2022) and Doan et al. (2023). Consequently, the paper is not ready yet for publication, and a stronger focus across these potential limitations would strengthen the paper significantly.

Xue et al. Imperceptible and multi-channel backdoor attack against deep neural networks. 2022.
Doan et al. Marksman backdoor: Backdoor attacks with arbitrary target class NeurIPS’23

**Reviewer Concerns:**

- WUb8
   - The core assumption of CSO is that the backdoor trigger introduces features that lie outside the normal feature subspace of the target class.
   - Quantifying “trigger–intrinsic” overlap (does CSO’s key assumption hold?)
   -  the effectiveness of the proposed defense on distribution-preserving backdoor attacks
- 236s
   - concern of adaptive attack using intrinsic feature as trigger
   - using intrinsic feature is not new
- FRMn:
   - Whether the model is backdoored or not is the main factor affecting the reverse engineering based detector.
   - Discrepancy between Neural Cleanse’s input space and feature space discussion in the paper.
   - when attack tries to reduce feature difference between benigh and and triggered samples, the attack may fail.
- VdYG
   - The writing is incoherent and hard to follow.
   - this method does not work well against backdoor features that are similar to intrinsic features.
   - The paper should evaluate other stealthy or clean label backdoor
- ERPo
   - dependent on clean data
   - effectiveness against clean-label and multi-trigger attacks remain unverified
   - adaptive results are underemphasized
- tvn1
   - How do CSO variant backdoor detectors depend on the number of images in the clean dataset per class?
   - behavior against adaptive attacker explicitly aligns trigger features with the target class’s intrinsic subspace
   - a way to efficiently tune this hyperparameter for different datasets and detectors?

Most of the concerns have been addressed well by the authors, making the paper stronger. However, the discussion on latent preservation attacks and multi-target/multi-trigger attacks is still limited, which should be expanded in the paper, as this demonstrates the cases where the method could potentially fail. For a backdoor work, I believe this is important.

**Reviewer Scores:**

One reviewer responded but kept the score. Other reviewers didn't much engage, and I'm not quite sure if they're willing to increase their scores on the paper.
    - WUb8: 6
    - 236s: 4
    - FRMn: 2
    - VdYG: 2
    - ERPo: 4
    - tvn1: 4

---

### Decision · Program_Chairs · 2026-01-26

Reject